# Ganglioside GT1b prevents selective spinal synapse removal following peripheral nerve injury

Jaesung Lee[1,2], Kyungchul Noh [1], Subeen Lee[3], Kwang Hwan Kim [4], Seohyun Chung[1], Hyoungsub Lim[1], Minkyu Hwang [4], Joon-Hyuk Lee[5], Won-Suk Chung [5], Sunghoe Chang [2✉] & Sung Joong Lee [1,3,4✉]

## Abstract

**After peripheral nerve injury, the structure of the spinal cord is actively regulated by glial cells, contributing to the chronicity of neuropathic pain. However, the mechanism by which peripheral nerve injury leads to synaptic imbalance remains elusive. Here, we use a pH-reporter system and find that nerve injury triggers a reorganization of excitatory synapses that is influenced by the accumulation of the ganglioside GT1b at afferent terminals. GT1b acts as a protective signal against nerve injury-induced spinal synapse elimination. Inhibition of GT1b-synthesis increases glial phagocytosis of excitatory pre-synapses and reduces excitatory synapses post-injury. In vitro analyses reveal a positive correlation between GT1b accumulation and the frequency of pre-synaptic calcium activity, with GT1b-mediated suppression of glial phagocytosis occurring through SYK dephosphorylation. Our study highlights GT1b's pivotal role in preventing synapse elimination after nerve injury and offers new insight into the molecular underpinning of activity-dependent synaptic stability and glial phagocytosis.**

**Keywords** Neuropathic Pain; Microglia; Astrocyte; Synapse Phagocytosis; GT1b
**Subject Category** Neuroscience

## Introduction

Neuropathic pain is chronic pathological pain caused by damage to or dysfunction of the nervous system (Colloca et al, 2017). It is increasingly clear that the activation of spinal cord glia plays a critical role in the development of nerve injury-induced neuropathic pain (Mika et al, 2013; Zhao et al, 2017). Nerve damage activates spinal microglia and astrocytes, which then release an array of pain-mediating molecules, including pro-inflammatory cytokines, brain-derived neurotrophic factor (BDNF), reactive oxygen species, and complements (Coull et al, 2005; Kim et al, 2004; Liu et al, 2017; Nie et al, 2013a). The cumulative effects of glial activation cause central sensitization in the spinal cord, which contributes to the development of allodynia and hyperalgesia (Carlton et al, 2009; Donnelly et al, 2020b). It is intriguing that central sensitization sometimes persists after the resolution of inflammatory spinal glial activation, resulting in chronic pain whose mechanisms remain elusive (Donnelly et al, 2020a; Ji et al, 2018).

Glial cells, particularly microglia and astrocytes, play pivotal roles in synaptic plasticity and remodeling by regulating both the formation and elimination of synapses (Allen and Lyons, 2018; Chung et al, 2015). They regulate synaptic formation by releasing synaptogenic molecules (Stogsdill and Eroglu, 2017). For instance, microglia and astrocytes foster excitatory synapse formation by releasing thrombospondin-4 and BDNF, respectively (Eroglu, 2009; Parkhurst et al, 2013; Yu et al, 2018). Meanwhile, synapses designated for removal are targeted by glial cells through "eat me" signals such as complements (C1q and C3), TNF-associated weak inducer of apoptosis, and phosphatidylserine (Cheadle et al, 2020; Hong et al, 2016; Schafer et al, 2012; Scott-Hewitt et al, 2020). Conversely, "don't eat me" signals, such as the interaction between CD47 and SIRPa, protect synapses from glial phagocytosis, preserving synaptic integrity during synaptic development and reorganization (Ding et al, 2021; Lehrman et al, 2018). A precise balance between "eat me" and "don't eat me" signals is essential to the integrity of neuronal circuits and the functional plasticity needed in a healthy central nervous system (Cornell et al, 2022; DeVries et al, 2024; Raiders et al, 2021). Disruptions in that balance can lead to aberrant synaptic loss or the preservation of unwanted synapses, both of which perpetuate pathological states (DeVries et al, 2024; Hong et al, 2016; Wilton et al, 2023). Given the pivotal roles of glia in synaptic reorganization, we hypothesized that nerve injury-induced glial activation would instigate synaptic reorganization of pain circuits at the spinal cord level, contributing to the chronicity of central sensitization.

As one mechanism of nerve injury-induced neuropathic pain, we previously showed that peripheral nerve injury induces the synthesis of the ganglioside GT1b in injured dorsal root ganglia (DRG) sensory neurons (Lim et al, 2020). That GT1b is

[1]Department of Neuroscience and Physiology, Dental Research Institute, School of Dentistry, Seoul National University, Seoul 08826, Republic of Korea. [2]Department of Physiology and Biomedical Sciences, Dementia Research Center, College of Medicine, Seoul National University, Seoul 08226, Republic of Korea. [3]Interdisciplinary Program in Neuroscience, College of Natural Science, Seoul National University, Seoul 08826, Republic of Korea. [4]Department of Brain and Cognitive Sciences, College of Natural Sciences, Seoul National University, Seoul 08826, Republic of Korea. [5]Department of Biological Sciences, Korea Advanced Institute of Science and Technology, Daejeon 34141, Republic of Korea. ✉E-mail: sunghoe@snu.ac.kr; sjlee87@snu.ac.kr

subsequently transported to afferent terminals in the spinal cord, where it induces central pain sensitization (Lee et al, 2021b; Lim et al, 2020). GT1b, a b-series ganglioside containing three sialic acid residues, can interact with sialic acid-binding immunoglobulin-type lectins (Siglecs) and transmembrane proteins containing immunoreceptor tyrosine-based inhibition motifs (ITIMs) on glial cells (Schnaar, 2023). Thus, signaling through Siglecs might trigger ITIM phosphorylation and subsequent SHP-1 activation, inhibiting glial phagocytic activity (Crocker et al, 2007; Macauley et al, 2014), suggesting that GT1b might serve as a potent "don't eat me" signal during synaptic reorganization following nerve injury.

For this study, we explored the role of glia in spinal synapse remodeling post-nerve injury, with a particular focus on the function of the ganglioside GT1b. We discovered that spinal glia selectively preserve excitatory pre-synapses depending on the presence of GT1b, which overrides complement-mediated synapse elimination. We confirmed the role of GT1b by demonstrating that inhibiting its synthesis led to increased glial synaptic elimination and consequent suppression of excitatory synaptic expansion. Our findings also reveal a direct correlation between GT1b accumulation on synaptic membranes and the frequency of pre-synaptic calcium activity. Additionally, we provide evidence of GT1b-mediated inhibition of glial phagocytosis. Our results identify GT1b as a potent and novel "don't eat me" signal that decisively orchestrates spinal synapse remodeling after nerve injury.

## Results and discussion

### Synapse type-specific and GT1b-dependent synapse elimination by spinal glia following peripheral nerve injury

To characterize the elimination of specific synapse types by spinal glia after nerve injury, we used a pH-reporter system to visualize four synaptic components: excitatory pre-synaptic (ExPre), excitatory post-synaptic (ExPost), inhibitory pre-synaptic (InhiPre), and inhibitory post-synaptic (InhiPost) (Fig. 1A,B). In this system, synaptic compartments that merely attached to or were engulfed by a glial membrane emitted both mCherry and enhanced green fluorescent protein (EGFP) fluorescence signals, whereas those in a phagolysosome emitted only the mCherry signal due to the different pKa of GFP and mCherry (Lee et al, 2021a). Following the expression of this system by injecting a virus into layers LI–II of the spinal cord dorsal horn (Fig. EV1A), we introduced nerve injury using the murine L5 spinal nerve transection (SNT) model, which is well-established for neuropathic pain, and validated pain induction with von-Frey and Hargreaves testing (Fig. EV2A–C). In this model, we observed an expansion of afferent terminals labeled by CGRP and an increase in excitatory synapses following nerve injury, whereas inhibitory synapses remained unchanged (Fig. EV3). Immunohistochemistry with microglia (Iba-1) and astrocyte (GFAP) markers, followed by 3D reconstruction using IMARIS, categorized the synaptic compartments as either attached/engulfed (yellow signal, representing green puncta in the 3D reconstructed image; herein referred to as attached) or entrapped in a phagolysosome (Red signal, representing red puncta in the 3D reconstructed image; herein referred to as phagocytosed or eliminated) by microglia or astrocytes (Movie EV1;

Figs. 1C and EV1B–E). Notably, in tissues expressing the pH-reporter targeted to ExPre, significant elimination by both microglia and astrocytes was observed both 3 and 7 days after nerve injury (Fig. 1C,D). Additionally, although no substantial changes were detected in the glial removal of ExPost compartments (Fig. 1E), a slight increase in the astrocytic elimination of InhiPre was noted 7 days post-SNT injury (dpi) (Fig. 1F), and their interactions with InhiPost were not significantly altered (Fig. 1G). To further assess microglial and astrocytic phagocytosis of InhiPre, we utilized VGAT and CD68 markers alongside Iba-1 and GFAP to quantify both glycinergic and GABAergic inhibitory synapses. These analyses revealed no significant phagocytosis of InhiPre by either microglia or astrocytes (Fig. EV4). Taken together, these data suggest that both microglia and astrocytes remodel excitatory synapses by phagocytic elimination of the pre-synaptic component in the spinal dorsal horn after nerve injury, with astrocytes having a role in eliminating inhibitory pre-synapses as well.

In a previous study, we reported that GT1b, a major ganglioside synthesized in the DRG, is transported and accumulates in spinal afferent terminals following peripheral nerve injury (Fig. EV5A,B) (Lim et al, 2020). Along with the transportation of GT1b, we observed a marked increase in GT1b within the excitatory pre-synaptic compartment, with 22.3% of VGLUT2 puncta co-localized with GT1b in the sham control, 35.8% at 3 dpi, and 47.6% at 7 dpi (Fig. 1H–J). Those data suggest that after SNT, spinal glia selectively trim synapses over time in a way that enriches GT1b-positive excitatory pre-synapses in the superficial layer of spinal dorsal horn.

Given those findings, we next focused on GT1b's functional role in synaptic elimination post-injury. It has been suggested that the sialic acid residues of gangliosides might interact with Siglecs on glial cells, thereby dampening the cells' phagocytic activity (Brown and Neher, 2014; Schnaar, 2023). That implies that GT1b might act as an inhibitory signal for glial phagocytosis. To test that hypothesis, we assessed whether glial synapse elimination was affected by the presence of GT1b in the synapse. When we counted microglia-phagocytosed ExPre using the pH-reporter system and GT1b immunostaining, we found that the microglial phagocytosis of GT1b-negative synapses increased significantly after injury (Fig. 1K,L). Astrocytes also preferentially eliminated GT1b-negative synapses, although that difference was not statistically significant between the pre- and post-injury conditions (Fig. 1M,N). These results collectively indicate that the accumulation of GT1b at pre-synaptic terminals appears to safeguard them against glial elimination during spinal synapse reorganization after nerve injury.

### Differential accumulation of complement and GT1b in phagocytosed and normal synaptic puncta

Complements orchestrate synaptic remodeling by phagocytic elimination during both developmental and pathological processes (Presumey et al, 2017; Wen et al, 2024). In the spinal cord, complement expression increases following peripheral nerve injury, and resolving those complements via antibodies or cobra venom factor has been shown to have analgesic effects (Levin et al, 2008; Nie et al, 2013b). Therefore, we hypothesized that the elimination of GT1b-negative ExPre synaptic compartments might be facilitated by complement activation. To explore that possibility,

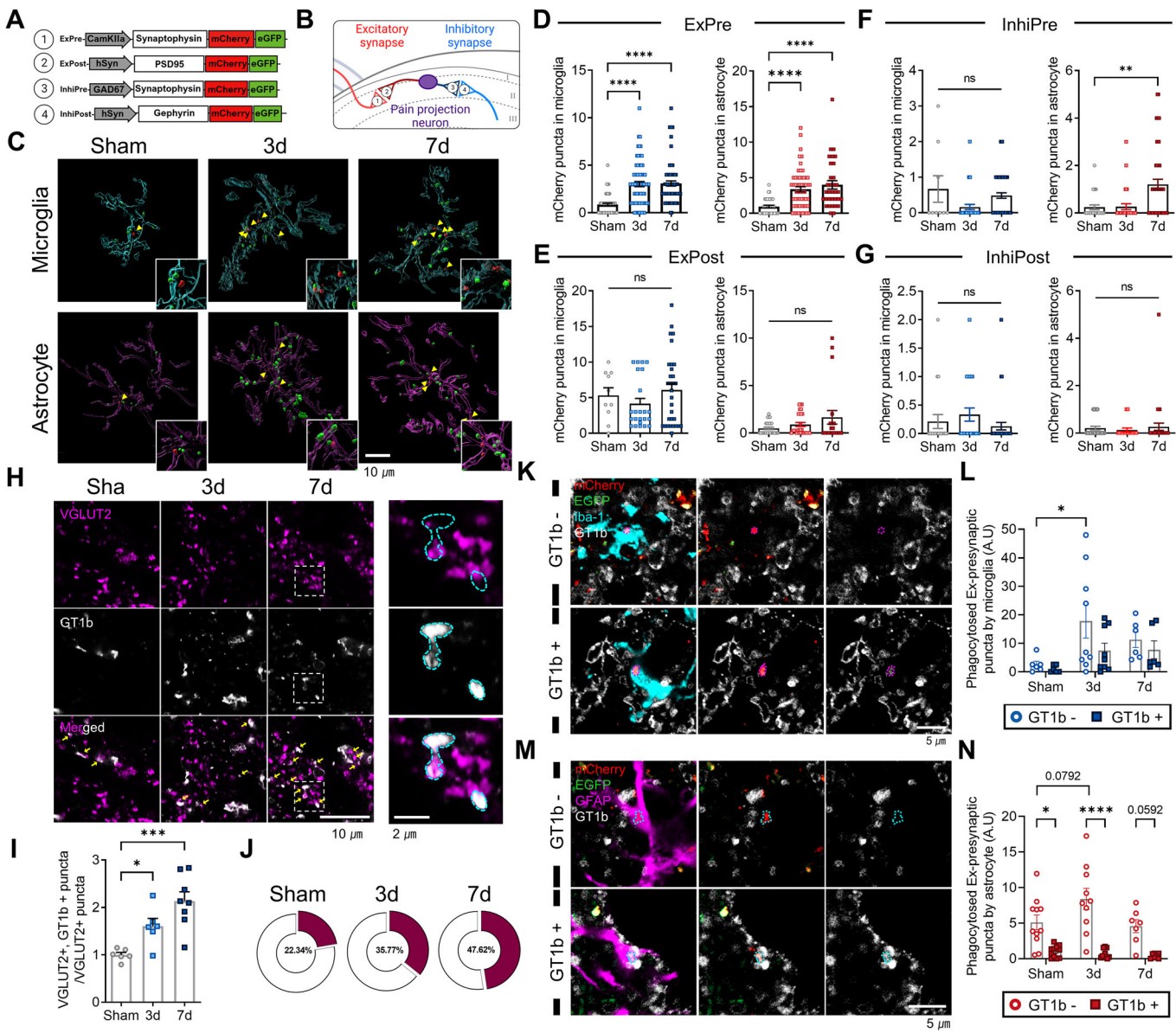

**Figure 1. Glial regulation of spinal synapses in the presence of GT1b following peripheral nerve injury.**

(A, B) Schematic illustrations depicting the four synaptic compartments targeted by the pH-reporter system for visualization: excitatory pre-synapse (ExPre), excitatory post-synapse (ExPost), inhibitory pre-synapse (InhiPre), and inhibitory post-synapse (InhiPost). (C) Immunohistochemical analysis using Iba-1 and GFAP in spinal cords expressing the ExPre pH-reporter. Arrowheads indicate ExPre puncta undergoing phagocytosis (red), and attached ExPre are indicated as green puncta. Scale bar = 10 μm. (D) Quantification of ExPre synapses undergoing phagocytosis by individual glial cells. P value (microglia); Sham vs. 3 d < 0.0001, Sham vs. 7 d < 0.0001. P value (astrocyte); Sham vs. 3 d < 0.0001, Sham vs. 7 d < 0.0001. n (microglia); Sham = 49, 3 d = 76, 7 d = 72. n (astrocyte); Sham = 38, 3 d = 57, 7 d = 33. (E) Quantification of ExPost synapses undergoing phagocytosis by individual glia. n (microglia); Sham = 10, 3 d = 23, 7 d = 31. n (astrocyte); Sham = 27, 3 d = 28, 7 d = 20. (F) Quantification of InhiPre synapses undergoing phagocytosis by individual glia. P value (microglia); Sham vs. 3 d = 0.05999, Sham vs. 7 d = 0.8177. P value (astrocyte); Sham vs. 3 d > 0.9999, Sham vs. 7 d = 0.0013. n (microglia); Sham = 9, 3 d = 37, 7 d = 66. n (astrocyte); Sham = 25, 3 d = 30, 7 d = 48. (G) Quantification of InhiPost synapses undergoing phagocytosis by individual glia. n (microglia); Sham = 19, 3 d = 24, 7 d = 39. n (astrocyte); Sham = 29, 3 d = 16, 7 d = 34. (H) Representative images of spinal cord immunostained for VGLUT2 and GT1b. Insets show a higher magnification of the selected areas. Scale bars = 10 and 2 μm. (I, J) Quantitative analysis of the proportion of GT1b + VGLUT2+ puncta. P value; Sham vs. 3 d = 0.0455, Sham vs. 7 d = 0.0003. n; Sham = 6, 3 d = 6, 7 d = 8. (K) Immunostained images of ExPre (identified with the pH-reporter) co-labeled with Iba-1 and GT1b. Scale bar = 5 μm. (L) Quantification of Iba-1+ microglia-mediated phagocytosis of ExPre synapses, differentiated by the presence or absence of GT1b. P value; Sham:GT1b− vs. 3 d:GT1b− = 0.0126. n; Sham = 6, 3 d = 9, 7 d = 6. (M) Immunostained images of ExPre (identified with the pH-reporter) co-labeled with GFAP and GT1b. Scale bar = 5 μm. (N) Quantification of GFAP+ astrocyte-mediated phagocytosis of ExPre synapses, differentiated by the presence or absence of GT1b. P value; Sham:GT1b- vs. Sham:GT1b + = 0.0103, 3 d:GT1b- vs. 3 d:GT1b + <0.0001. n; Sham = 11, 3 d = 10, 7 d = 7. Data are presented as the mean ± SEM; *P < 0.05, **P < 0.01, ***P < 0.001, ****P < 0.0001; one-way ANOVA with Dunnett's multiple comparison test and two-way ANOVA with Tukey's and Dunnett's multiple comparison test. See also Movie EV1. Source data are available online for this figure.

we stained for C3, a major effector in complement-mediated synapse elimination, in spinal cord expressing the ExPre pH-reporter. Our observations revealed that complement co-localization with phagocytosed synaptic puncta increased significantly after nerve injury (Fig. 2A,B). Notably, phagocytosed synaptic puncta predominantly co-localized with C3, whereas GT1b alone-positive phagocytosed ExPre were absent (Fig. 2C,D). In contrast, normal synaptic puncta exhibited an increased ratio of puncta positive for GT1b alone or GT1b and C3 together following nerve injury (Fig. 2C,D).

To elucidate the relative roles of C3 and GT1b in the process of synapse elimination, we used the fluorescence intensity of EGFP to track their deposition across different synaptic compartments. We categorized synapses based on mCherry puncta co-localizing with varying percentages of EGFP, creating synapse categories of less than 10%, 10–25%, 25–50%, 50–75%, and more than 75% EGFP co-localization. Our results reveal that after nerve injury, C3 deposition increased markedly across almost all ExPre synapses, irrespective of the EGFP intensity levels (Fig. EV6A). These data suggest that C3-postitive ExPre synapses are evenly distributed between glial membrane-attached and phagocytosed puncta. In contrast, GT1b deposition increased predominantly in synapses with higher EGFP levels (Fig. EV6B), indicating GT1b's preferential localization in the glial membrane-attached rather than phagocytosed synaptic puncta. This differential pattern of deposition suggests that although the complement system generally promotes synaptic pruning, GT1b likely acts as a protective modulator that preserves synaptic integrity by overriding complement-mediated synapse elimination mechanisms.

### Suppression of GT1b synthesis in injured sensory neurons enhances excitatory synaptic phagocytosis by glia, which ameliorates excitatory synaptic reorganization in the spinal cord

Given the putative role of GT1b in protecting ExPre, we investigated how suppressing GT1b synthesis affects synaptic reorganization. We used D-threo-1-phenyl-2-decanoylamino-3-morpholino-1-propanol·HCl (PDMP), a pharmacological inhibitor of ganglioside synthesis, to reduce spinal GT1b levels (Lim et al, 2020; Vunnam and Radin, 1980). Intrathecal PDMP administration significantly lowered GT1b levels in the DRG (Fig. EV7A,B) and spinal dorsal horn (Fig. EV7C) at both the basal level (sham) and after SNT injury, validating our method (Fig. EV7). Subsequently, we observed an increase in glial synaptic elimination, suggesting that a lack of GT1b boosts glial phagocytosis of ExPre compartments (Fig. 3A–C). Furthermore, we assessed whether inhibiting GT1b synthesis affected excitatory synapse remodeling after peripheral nerve injury. We found that PDMP administration ameliorated injury-induced excitatory synapse expansion (Fig. 3D–F). We then used electrophysiology to assess the effects of GT1b depletion on synaptic transmission. SNT induced a frequency increase in the spontaneous excitatory post-synaptic current (sEPSC) in neurons from the superficial layers (LI–II) on the ipsilateral side but not the contralateral side (Fig. 3G–I). Notably, PDMP administration almost completely abrogated the SNT-induced sEPSC frequency increase on the ipsilateral side (Fig. 3G–I). To exclude putative pharmacological side effects of PDMP, we genetically blocked GT1b synthesis. GT1b is

synthesized from GD1b by the ST3Gal-2 or -3 enzyme (Sturgill et al, 2012). We found that the GT1b level was completely reduced in the spinal cords of St3gal-2/3 double knockout mice (Fig. EV7D). Furthermore, the SNT-induced increase in excitatory synapses was almost completely abrogated in the St3gal-2/3 double knockout mice (Fig. 3J–L). To exclude the contribution of GT1b from spinal cord neurons, we used Scn10a-Cre/St3gal2f/f mice, in which the St3gal2 gene is deleted only in sensory neurons expressing Cre under the Na$_v$1.8 promoter (Agarwal et al, 2004). In our previous study, we found that SNT increased GT1b in afferent sensory axons mainly via St3gal2, not St3gal3 (Lim et al, 2020). In that way, the nerve injury-induced GT1b increase and subsequent central pain sensitization were attenuated in Scn10a-Cre/St3gal2f/f mice (Lim et al, 2020). In line with our previous report, SNT-induced GT1b expression in the spinal dorsal horn was significantly ameliorated in the Scn10a-Cre/St3gal2f/f mice (Fig. EV7E). Notably, sensory neuron-specific St3gal2 deletion completely suppressed SNT-induced excitatory synapse expansion (Fig. 3M–O). Taken together, these data indicate that a lack of SNT-induced GT1b synthesis in afferent sensory neurons leads to excessive glial elimination of excitatory synapses, which subsequently attenuates the injury-induced increase in excitatory synapses and sEPSC frequency.

### GT1b accumulation in the pre-synaptic compartment correlates with synaptic calcium activity

Glia-mediated synapse elimination occurs not just during development, but also under both physiological and pathological conditions in a synaptic activity-dependent manner (Jung and Chung, 2018; Yasuda et al, 2021). Therefore, we conducted further in vitro experiments to test the relationship between GT1b accumulation at synaptic membranes and the level of synaptic activity. Primary DRG sensory neurons were infected with AAV-hSyn1-SYP1-GCaMP7f and labeled with CF647-conjugated GT1b antibodies (Fig. 4A,B). Spontaneous calcium transients (sCaTs) at pre-synaptic boutons were analyzed at GT1b-positive and GT1b-negative synapses (Fig. 4C; Movie EV2). Interestingly, GT1b-labeled pre-synaptic sCaTs were characterized by lower amplitudes and higher frequencies than their GT1b-negative counterparts (Figs. 4D and EV8A,B). Further analysis using multiple linear regression revealed a stark contrast in the activity profiles of the two synapse groups (Fig. 4E). These observations support the possibility that GT1b accumulation at the synaptic membrane is associated with synaptic activity levels, potentially serving as a mechanistic basis for selectively preserving active synapses during glial-mediated synaptic remodeling.

### GT1b inhibits the phagocytic activity of glial cells via SYK dephosphorylation

CD47 inhibits microglial phagocytosis by activating SIRPα, which might facilitate SHP-SYK signaling pathways (Bian et al, 2016; Lehrman et al, 2018; Zhang et al, 2015). We hypothesized that GT1b might similarly deliver a "don't eat me" signal to glia to protect synapses. To test that hypothesis, we conducted a phagocytosis assay in mixed glia and BV2 microglia using live cell imaging and flow cytometry (Fig. 5A,B). In primary mixed glia, GT1b treatment reduced the glial phagocytosis of pH-sensitive

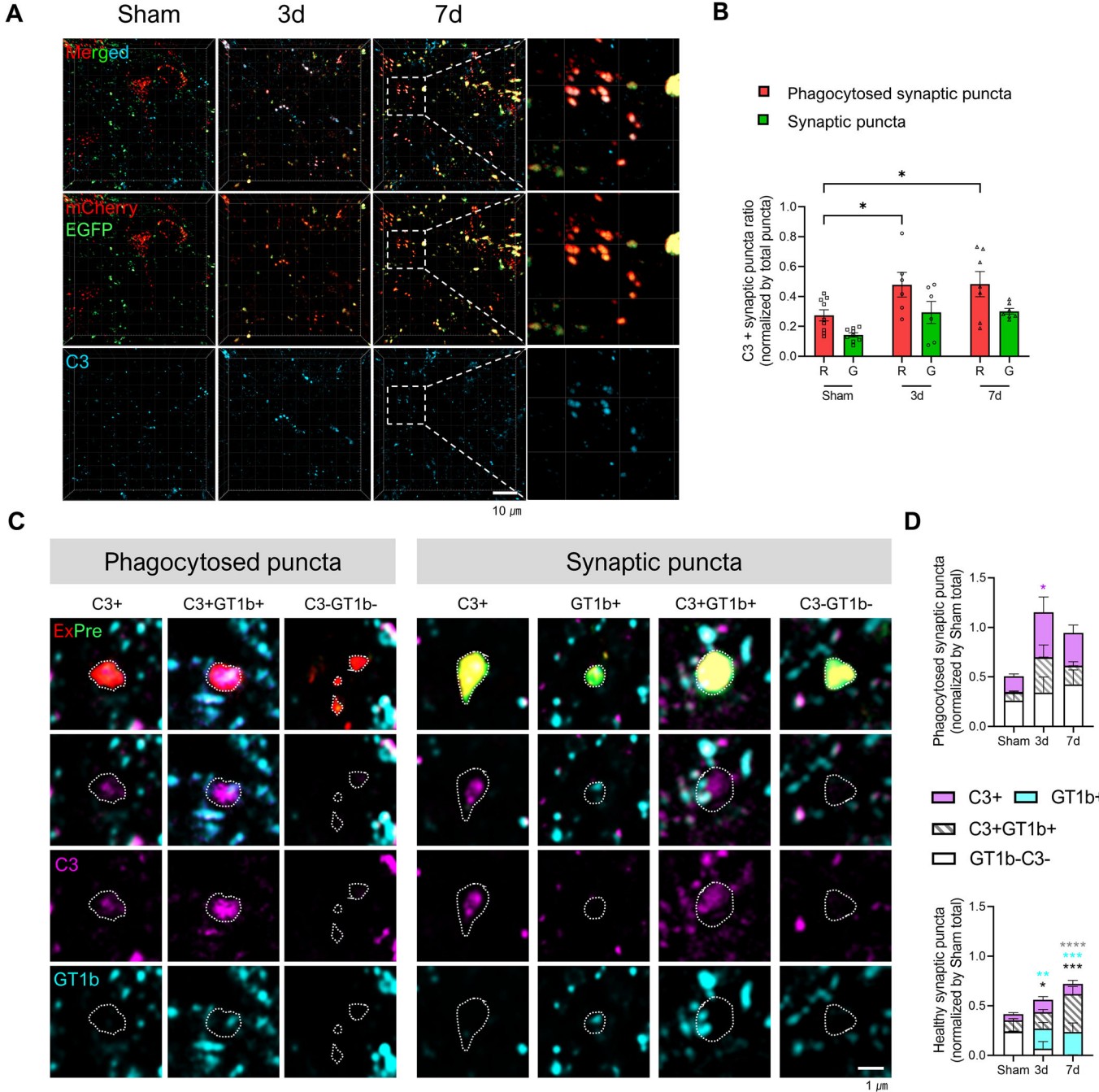

**Figure 2. Complement and GT1b mediate opposing functions in spinal synapse reorganization.**

(A) Representative image of C3 staining in ExPre pH-reporter-expressing spinal cord tissue following peripheral nerve injury. Scale bar = 10 μm. (B) The ratio of C3-positive synaptic puncta normalized by the total synaptic puncta in each sample. Synaptic puncta are categorized into phagocytosed synaptic puncta and non-phagocytosed synaptic puncta based on their co-localization with EGFP. P value (Phagocytosed puncta); Sham vs. 3 d = 0.0203, Phagocytosed puncta: Sham vs. 7 d = 0.013. P value (non-phagocytosed puncta); Sham vs. 3 d = 0.1016, Sham vs. 7 d = 0.0665. n; sham = 9, 3 d = 6, 7 d = 7. (C) Double staining of GT1b and C3 in ExPre-expressing spinal cord tissue. Phagocytosed puncta are grouped into C3-positive, C3-/GT1b-double positive, and both negative categories, and normal synaptic puncta are categorized as C3-positive, Gt1b-positive, both positive, and both negative. Magnified images are indicated by dashed boxes. Scale bar = 1 μm. (D) Quantification of the changes in C3 and GT1b co-localization in phagocytosed and normal synaptic puncta following nerve injury. P value (Phagocytosed puncta); Sham:C3+ vs. 3 d:C3 + = 0.0374. P value (Normal puncta:C3 + GT1b + ); Sham vs. 7 d < 0.0001. P value (Normal puncta:GT1b + ); Sham vs. 3 d = 0.0037, Sham vs. 3 d = 0.0008. P value (Normal puncta:GT1b-C3-); Sham vs. 3 d = 0.0182, Sham vs. 7 d = 0.0003. n; Sham = 8, 3 d = 6, 7 d = 8. Data are presented as the mean ± SEM; *P < 0.05, **P < 0.01, ***P < 0.001, ****P < 0.0001; two-way ANOVA with Dunnett's multiple comparison test. See also Fig. EV6. Source data are available online for this figure.

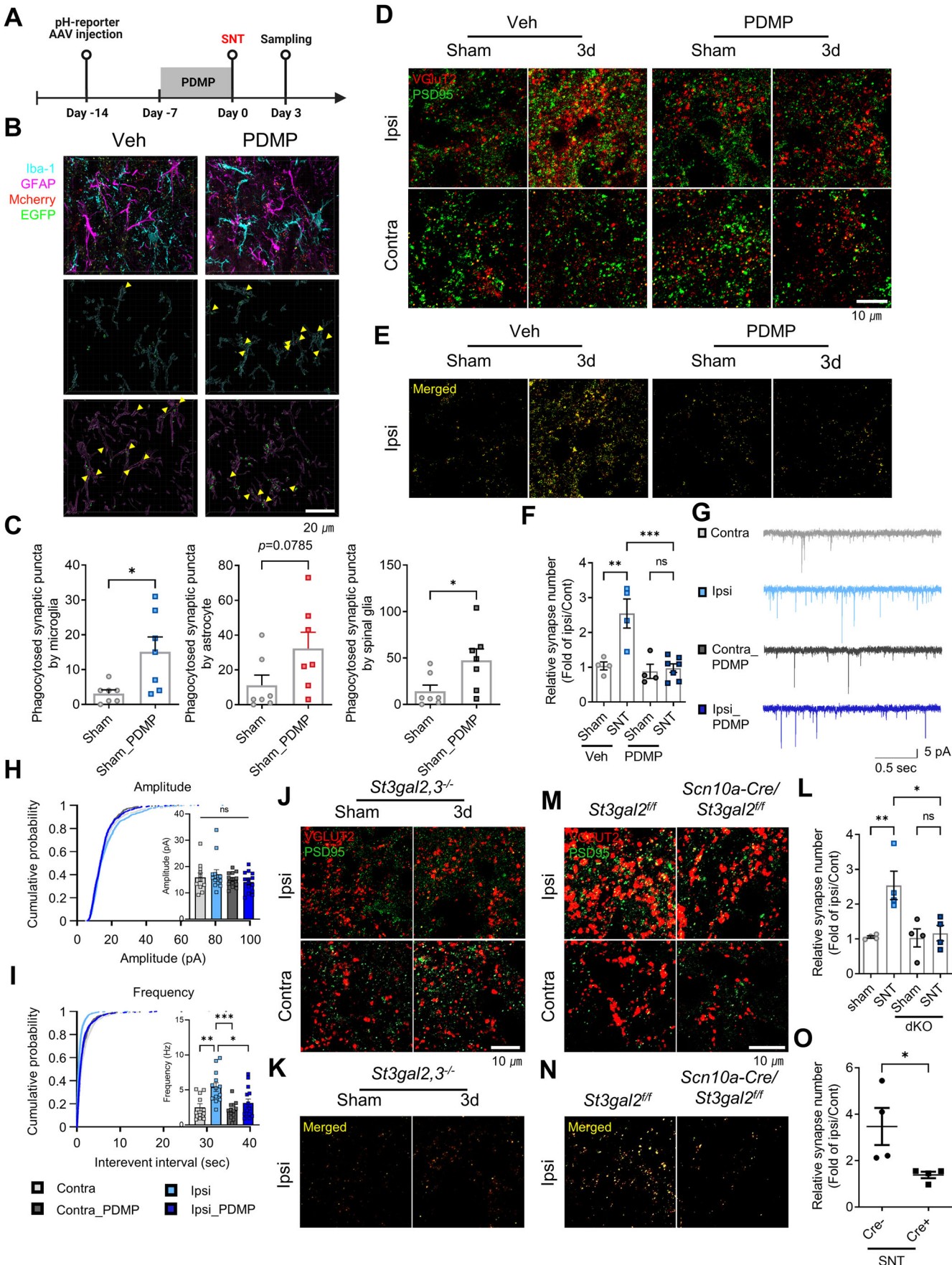

**Figure 3. Inhibition of GT1b synthesis enhances glial phagocytosis and disrupts excitatory synaptic expansion.**

(A) Experimental timeline of PDMP treatment to suppress GT1b synthesis. (B) Representative images of ExPre (identified with the pH-reporter) in the spinal cord immunostained with Iba-1 and GFAP. Scale bar = 20 μm. Yellow arrowheads represent phagocytosed ExPre puncta. (C) Quantification of glial phagocytosis of synapses with and without PDMP treatment. *P* value (microglia); Sham vs Sham_PDMP = 0.017. *P* value (astrocyte); Sham vs Sham_PDMP = 0.0785. *P* value (gali); Sham vs. Sham_PDMP = 0.0351. *n*; Sham = 7, Sham_PDMP = 7. (D) Immunostaining for VGLUT2 and PSD95 to visualize excitatory synapses. Scale bar = 10 μm. The co-localized excitatory synaptic puncta are shown in (E). (F) Analysis of relative excitatory synapse numbers. *P* value; Veh_Sham vs. Veh_SNT = 0.0022, Veh_SNT vs. PDMP_SNT = 0.0005. *n*; Veh_Sham = 4, Veh_SNT = 4, PDMP_Sham = 4, PDMP_SNT = 7. (G) Sample traces of spontaneous excitatory post-synaptic currents (sEPSCs) from the ipsilateral and contralateral sides. (H, I) Changes in sEPSC amplitude and frequency. *P* value; SNT_Ipsi vs. SNT_Contra = 0.0028, SNT_Ipsi vs. PDMP:SNT_Ipsi = 0.0003, SNT_Ipsi vs. PDMP:SNT_Ipsi = 0.145. *n*; SNT_Ipsi = 13, SNT_Contra = 11, PDMP:SNT_Ipsi = 14, PDMP:SNT_Contra = 13. (J) Immunostaining of spinal cords from *St3gal2/3* knockout mice with VGLUT2 and PSD95 and (K) their co-localized excitatory synapses. Scale bar = 10 μm. (L) Quantification of inhibitory synaptic puncta co-localized after SNT. *n*; Sham = 4, SNT = 4, dKO_Sham = 4, dKO_SNT = 4. (M) Immunostaining of spinal cords from *Scn10a-Cre/St3gal2^{f/f}* mice with VGLUT2 and PSD95 and (N) their co-localized excitatory synapses. *P* value; Sham vs SNT = 0.0093, SNT vs. dKO:SNT = 0.0157. Scale bar = 10 μm. (O) Quantification of inhibitory synaptic puncta co-localized after SNT. *P* value; Cre- vs Cre + = 0.0351. *n*; Cre- = 4, Cre + = 4. Data are presented as the mean ± SEM; *P < 0.05, **P < 0.01, ***P < 0.001, ****P < 0.0001; one-way ANOVA with Tukey's multiple comparison test and two-way ANOVA with Tukey's multiple comparison test. See also Fig. EV7. Veh vehicle control for PDMP, Ipsi ipsilateral side, Contra contralateral side, SNT L5 spinal nerve transection model for neuropathic pain, Sham sham control for SNT, dKO double KO of St3gal2/3, Cre Scn10a-Cre. Source data are available online for this figure.

pHrodo Red *E. coli* bioparticles in a dose-dependent manner (Fig. 5C,D). That inhibitory effect was further confirmed by treating BV2 microglial cells with GT1b, which decreased the phagocytosis of pHrodo bioparticles, as reflected in both the mean fluorescent intensity and quantity of pHrodo-positive cells (Fig. 5E,F). Moreover, in a primary mixed glial culture, GT1b treatment resulted in a significant reduction in the phagocytic activity of both microglia and astrocytes (Fig. 5G,H). Further investigation into the molecular underpinnings of those results revealed that GT1b treatment leads to the dephosphorylation of SYK in primary mixed glia (Fig. 5I,J), aligning with the theory that GT1b interaction with its receptor Siglecs on glia triggers SYK dephosphorylation and subsequent inhibition of glial phagocytosis. Taken together, these data indicate that GT1b accumulates preferentially in active pre-synaptic membranes, where it delivers a "don't eat me" signal to spinal cord glia, enriching excitatory synapses in the dorsal horn after nerve injury, and potentially contributing to the chronicization of central pain sensitization (Fig. EV9).

In this study, we unveiled the function of GT1b as a protective "don't eat me" signal within spinal synapses, offering a novel insight into its synapse-preserving functions within pain circuits. This discovery not only advances our understanding of synaptic remodeling post-nerve injury, but also identifies GT1b as a potential therapeutic target for mitigating chronic neuropathic pain.

Using a pH-reporter system, we observed glial involvement in the type-specific elimination of synapses, with both microglia and astrocytes actively participating in the removal of ExPre synapses and astrocytes also targeting InhiPre compartments during the later stages of neuropathy. This pattern of selective elimination could be attributed to differential levels of synaptic attachment to glial cells. According to our analysis with the pH-reporter system (Fig. EV1), ExPre terminals exhibit a low attachment ratio to astrocytes and microglia, suggesting lower structural stability, whereas ExPost and InhiPost show higher attachment ratios, implying greater stability. Notably, InhiPre terminals exhibit a gradual decrease in their attachment ratio after nerve injury, which highlights their increased vulnerability to glial phagocytosis in the later stages of nerve injury.

Previous reports have demonstrated that peripheral nerve injury leads to a loss of inhibitory synapses, thereby increasing excitatory input to the spinal dorsal horn (Xu et al, 2023; Yousefpour et al,

2023). However, in our spinal nerve transection (SNT) injury model, we did not observe a significant change in the total number of inhibitory synapses, despite evidence of astrocytic phagocytosis of GAD-promoter-expressed inhibitory pre-synapses (InhiPre). These differential patterns of spinal synapse elimination suggest that variations in the injury mode, pain onset, and the examined spinal cord region may contribute to distinct structural reorganization patterns. Another possibility is that the extent of inhibitory synapse elimination in the late post-injury phase may vary depending on the nerve injury model. In our study, we utilized the GAD promoter to label inhibitory pre-synapses, which specifically marks GABAergic pre-synaptic terminals. To include glycinergic inhibitory pre-synapses as well, we conducted triple staining for VGAT, CD68, and Iba-1/GFAP to quantify the phagocytosed inhibitory pre-synapses following peripheral nerve injury. Our analysis revealed no significant differences in inhibitory synapse elimination, suggesting that glycinergic synapses may not be selectively targeted for phagocytosis under these conditions.

Our findings further refine the understanding of complement-mediated synapse elimination. A critical gap in the previous model was its failure to explain how specific synapses are selectively targeted for removal. Our data show that although the sham control presented low levels of complement or GT1b co-localization at all synapses, nearly all ExPre synapses exhibited complement co-localization post-injury, with phagocytosed synapses notably lacking GT1b co-localization. This observation supports our hypothesis that although all ExPre synapses could be vulnerable to elimination, those that accumulate GT1b post-injury survive, escaping complement-mediated synapse elimination.

Moreover, we demonstrated that GT1b accumulation correlates with synaptic calcium activity in vitro, suggesting an activity-dependent mechanism of synapse protection. This relationship is crucial for understanding how synaptic activity influences glial responses, particularly in the context of heightened nociceptive signal transmission during neuropathic pain. Interestingly, recent studies have shown that synapses with persistent calcium accumulations are preferentially targeted for elimination (Jafari et al, 2021). This process may involve local cytoskeletal destabilization, mediated by the activation of calcium-dependent proteases or phosphatases, which ultimately leads to synapse elimination (Andres et al, 2013; Kuchibhotla et al, 2008). Although the direct

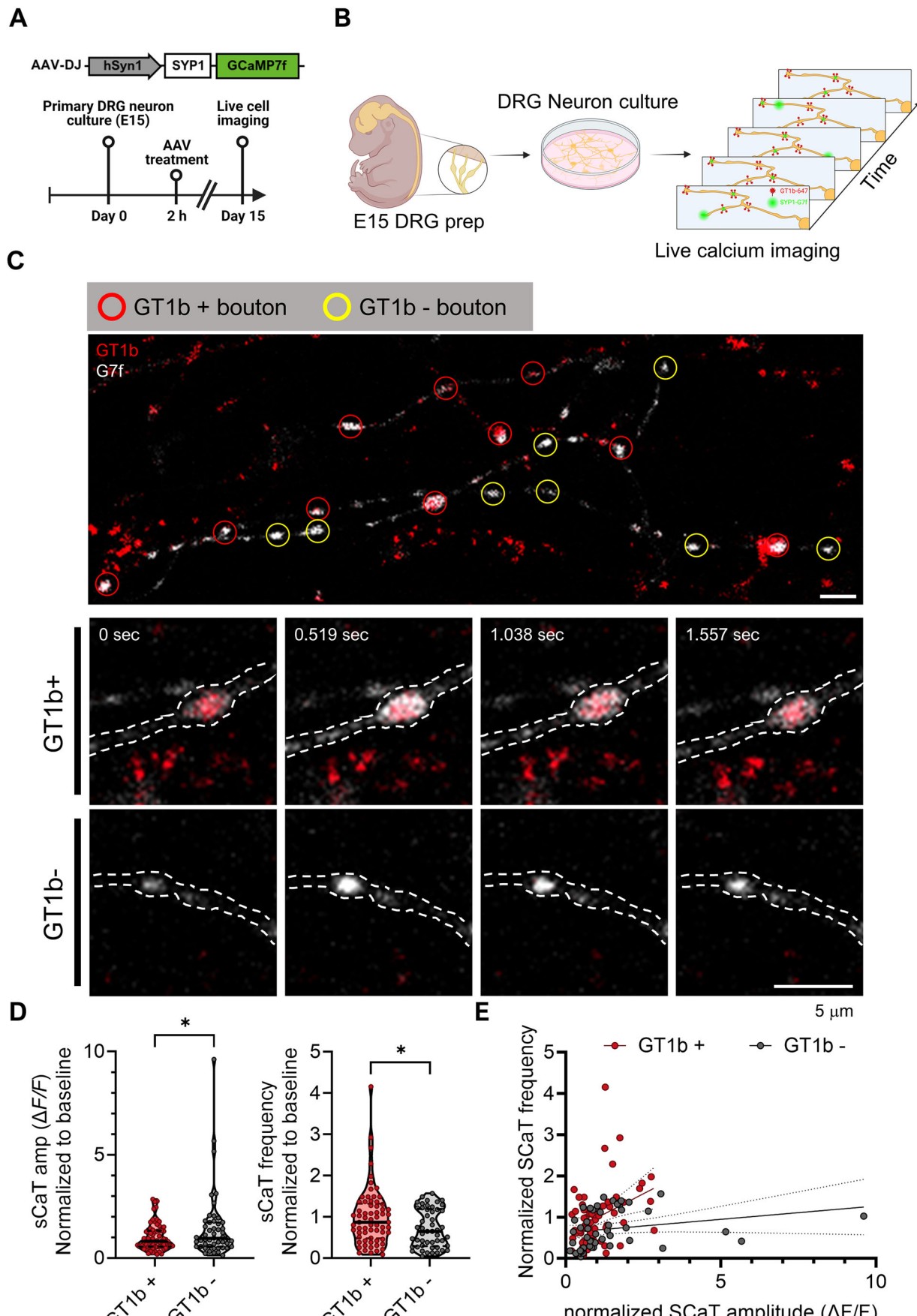

Figure 4. **Correlation between pre-synaptic calcium activity and GT1b accumulation.**

(A) Experimental timeline of in vitro synaptic activity measurement. (B) Schematic outlining primary sensory neuron culture and subsequent live cell imaging to examine the association between GT1b accumulation at pre-synaptic sites and spontaneous calcium transients (sCaTs). (C) Representative images of live sensory neurons tagged with SYP1-GCaMP7f, alongside GT1b labeling. Axonal fibers are highlighted with dashed lines. Scale bar = 5 μm. (D) Amplitude and frequency of sCaTs. (E) Grouped data comparison of the frequency and amplitude of sCaTs differentiated by GT1b accumulation. $P$ value (amplitude); GT1b+ vs. GT1b− = 0.0488. $P$ value (Frequency); GT1b+ vs. GT1b− = 0.021. $n$; GT1b− = 50, GT1b + = 66. Data are presented as mean ± SEM; *$P < 0.05$; Student's $t$ test and simple linear regression. See also Fig. EV8 and Movie EV2. Source data are available online for this figure.

causality between GT1b accumulation and synaptic activity requires further exploration, our results imply that GT1b can protect specific synapses in an activity-dependent manner.

Glial phagocytosis, a primary mechanism for synaptic elimination, can be inhibited by "don't eat me" signals (Brown and Neher, 2014; Elward and Gasque, 2003). Our findings indicate that GT1b suppresses glial phagocytosis, supporting its role as a novel "don't eat me" signal that contributes to synapse protection. The precise mechanism underlying the GT1b-mediated inhibition of phagocytosis likely involves the activation of ITIM domains through interactions with Siglecs (Siddiqui et al, 2019). That activation could, in turn, lead to the dephosphorylation of key regulators of DAP12-dependent phagocytosis, such as SYK, via the action of SHP-1 (Linnartz-Gerlach et al, 2014; Peng et al, 2010; Turnbull and Colonna, 2007). In further experiments, we will aim to verify this pathway, providing further insights into the molecular interactions that protect synapses from glial phagocytosis.

In conclusion, this study demonstrates that, following nerve injury, structural remodeling of pain circuitry, particularly between injured afferent and pain projection neurons, is heavily governed by the interaction between GT1b and spinal glia. GT1b, accumulating in an activity-dependent manner, plays a pivotal role in selectively preserving active spinal synapses from nerve injury-induced spinal synapse elimination by suppressing glial phagocytosis. Thus, we propose that GT1b serves as a novel "don't eat me" signal in spinal synapse reorganization and could be a possible therapeutic target for managing chronic neuropathic pain. This finding underscores the importance of GT1b in modulating synaptic stability and highlights its potential as a novel intervention point in the treatment of neuropathic pain.

## Methods

### Reagents and tools table

| Reagent/resource | Reference or source | Identifier or catalog number |
|---|---|---|
| **Experimental models** | | |
| Mouse: C56BL/6J | DBL | C57BL/6NTac |
| Mouse: *St3gal2/3 -/-* | Sturgill et al, 2012 | N/A |
| Mouse: *Scn10a-Cre/St3gal2^{f/f}* | Agarwal et al, 2004 | N/A |
| BV2 cell line | Accegen | Cat # ABV-TC212S |
| **Recombinant DNA** | | |
| pAAV-hSyn1-SYP1-GCaMP7f-P2A-ECFP | Subcloned | |

| Reagent/resource | Reference or source | Identifier or catalog number |
|---|---|---|
| pAAV-retro2-CaMKIIa-hSyn-mCherry-eGFP | Subcloned | |
| pAAV-GAD67-synaptophysin-mCherry-eGFP- | Lee et al, 2021a | |
| pAAV-hSyn-PSD95-mCherry-eGFP | Lee et al, 2021a | |
| pAAV-hSyn-Gephrin-mCherry-eGFP | Lee et al, 2021a | |
| **Antibodies** | | |
| Mouse monoclonal Anti-CGRP antibody | Abcam | Cat# ab81887 |
| Mouse monoclonal Anti-VGLUT2 antibody | Abcam | Cat# ab79157 |
| Rabbit monoclonal Anti-VGLUT2 antibody | Abcam | Cat# ab216463 |
| Rabbit polyclonal Anti-PSD95 antibody | Invitrogen | Cat# 51-6900 |
| Rabbit polyclonal Anti-VGAT antibody | Synaptic System | Cat# 131 006 |
| Mouse monoclonal Anti-Gephyrin antibody | Synaptic System | Cat# 147 011 |
| Rabbit polyclonal Anti-Iba1 antibody | Wako | Cat# 019-19741 |
| Mouse monoclonal Anti-Iba1 antibody | Invitrogen | Cat# MA5-27726 |
| Mouse monoclonal Anti-GFAP antibody | Sigma-Aldrich | Cat# MAB360 |
| Rabbit polyclonal Anti-GFAP antibody | DAKO | Cat# Z0334 |
| Mouse monoclonal Anti-GT1b antibody | Sigma-Aldrich | Cat# MAB5608 |
| Rat monoclonal Anti-C3 antibody | Abcam | Cat# ab11862 |
| Isolectin Gs-IB4, Alexa Fluor 647 conjugate | Invitrogen | Cat# I32450 |
| Mouse monoclonal Anti-ASPA antibody | Santa Cruz Biotech | Cat# sc-377308 |
| Rat monoclonal Anti-CD31 antibody | Invitrogen | Cat# 14-0311-82 |
| Brilliant violet 421 Anti-mouse/human CD11b antibody | BioLegend | Cat# 101235 ACSA |
| Anti-mouse ACSA-2-APC antibody | Miltenyi Biotec | Cat# 130-102-315 |

| Reagent/resource | Reference or source | Identifier or catalog number |
|---|---|---|
| TruStain FcX (anti-mouse CD16/32) antibody | BioLegend | Cat # 101319 |
| Phospho-SYK (Tyr525/526) antibody | Cell Signaling | Cat# 2710T |
| SYK antibody | Cell Signaling | Cat# 2712T |
| Rabbit monoclonal GAPDH-antibody | Cell Signaling | Cat # 5174S |
| Goat anti-rabbit IgG (H + L) Secondary antibody, Fluorescein (FITC) AffiniPure™ | Jackson ImmunoResearch | Cat # 111-095-003 |
| Goat anti-rabbit IgG (H + L) Secondary antibody, Alexa Fluor® 647 AffiniPure™ | Jackson ImmunoResearch | Cat # 111-095-003 |
| Goat anti-mouse IgG (H + L) Secondary antibody, Fluorescein (FITC) AffiniPure™ | Jackson ImmunoResearch | Cat # 115-095-003 |
| Goat anti-rat IgG (H + L) Secondary antibody, Cy™3 AffiniPure™ | Jackson ImmunoResearch | Cat # 112-165-143 |
| Goat anti-mouse IgG (H + L) Secondary antibody, Cy™3 AffiniPure™ | Jackson ImmunoResearch | Cat # 115-165-003 |
| Goat anti-rat IgG (H + L) Secondary antibody, Fluorescein (FITC) AffiniPure™ | Jackson ImmunoResearch | Cat # 112-095-020 |
| Donkey anti-mouse IgG (H + L) Secondary antibody, Fluorescein (FITC) AffiniPure™ | Jackson ImmunoResearch | Cat # 715-095-151 |
| Donkey anti-mouse IgG (H + L) Secondary antibody, Alexa Fluor® 647 AffiniPure™ | Jackson ImmunoResearch | Cat # 715-605-150 |
| Donkey anti-mouse IgG (H + L) Secondary antibody, Cy™3 AffiniPure™ | Jackson ImmunoResearch | Cat # 715-165-150 |
| Donkey anti-rabbit IgG (H + L) Secondary antibody, Cy™3 AffiniPure™ | Jackson ImmunoResearch | Cat # 711-165-152 |
| Donkey anti-rabbit IgG (H + L) Secondary antibody, Alexa Fluor® 647 AffiniPure™ | Jackson ImmunoResearch | Cat # 711-605-152 |
| Donkey anti-rabbit IgG (H + L) Secondary antibody, Fluorescein (FITC) AffiniPure™ | Jackson ImmunoResearch | Cat # 711-095-152 |
| Goat anti-mouse IgG (H + L) Secondary antibody, Alexa Fluor™ 405 | Invitrogen | Cat # A-31553 |
| Goat anti-rabbit IgG (H + L)-HRP | Abfrontier | Cat # LF-Sa8002 |

| Reagent/resource | Reference or source | Identifier or catalog number |
|---|---|---|
| Goat anti-mouse IgG -HRP 0.5 ml | Abfrontier | Cat # LF-SA8001 |
| **Critical commercial assays** | | |
| Pierce™ BCA protein assay kit | Thermo Scientific™ | Cat # 23227 |
| Mix-n-Stain CF647 antibody labeling kit | Sigma-Aldrich | Cat # MX647s100 |
| **Chemicals, enzymes and other reagents** | | |
| FITC-conjugated isolectin B4 | Sigma-Aldrich | Cat # L2895 |
| 1 M Tris-HCl, pH 6.8 | Biosesang | Cat # TR2016-050-68 |
| 1.5 M Tris-HCl, pH 8.8 | Biosesang | Cat # 4020-050-88 |
| TEMED (N,N,N′,N′-tetramethylethylenediamine) | Sigma-Aldrich | Cat # 1002522712 |
| Ammonium persulfate (APS) | DUKSAN | Cat # 7727-54-0 |
| 10% Ammonium persulfate (APS) | Biosesang | Cat # AC2013-110-00 |
| 5X SDS-PAGE loading buffer | Intronbio | Cat # IBS-BS002 |
| RIPA buffer | LPS solution | Cat # CBR002 |
| 30% Acrylamide/Bis solution 29:1 | Bio-Rad | Cat # 1610156 |
| WEST save gold (ECL solution) | Abfrontier | Cat # LF-QC0103 |
| PageRuler Plus prestained protein ladder | Thermo Scientific™ | Cat # 26619 |
| HEPES | Gibco | Cat # 15630-080 |
| GlutaMAX | Gibco | Cat # 35050-061 |
| Collagenase P | Roche | Cat # 11 213 865 001 |
| MEM NEAA | Gibco | Cat # 11140-050 |
| Dispase | Gibco | Cat # 17105-041 |
| Sodium pyruvate | Gibco | Cat # 11360-070 |
| Collagenase A | Roche | Cat # 10103578001 |
| pHrodo™ BioParticles™ conjugates for phagocytosis and phagocytosis kit for flow cytometry | Invitrogen | Cat # P35361 |
| Tyrode's solution | Sigma-Aldrich | Cat # T1788 |
| DMEM high glucose | Welgene | Cat # LM 001-01 |
| HBSS | Welgene | Cat # LB 003-02 |
| 0.25% Trypsin-EDTA | Gibco | Cat # 25200-056 |
| Anti-Anti (100X) | Gibco | Cat # 15240-062 |
| L-Glutamine 200 mM (100X) | Gibco | Cat # 25030-081 |
| Ganglioside GT1b (bovine) (sodium salt) | Matreya (Cayman) | Cat # 15588 |
| D-threeo-PDMP | Matreya (Cayman) | Cat # 1756 |
| Formaldehyde solution | Sigma-Aldrich | Cat # 47608-1L-F |
| Urethane | Sigma-Aldrich | Cat # U2500-500G |
| Sodium dihydrogen phosphate dihydrate | JUNSEI | Cat # 84190-0350 |
| D-Glucose | JUNSEI | Cat # 64220S0601 |

| Reagent/resource | Reference or source | Identifier or catalog number |
|---|---|---|
| Sodium hydrogen carbonate | JUNSEI | Cat # 43305-0301 |
| Calcium chloride | JUNSEI | Cat # 18235-0301 |
| Magnesium chloride hexahydrate | JUNSEI | Cat # 19275S0301 |
| Sodium chloride ≥99.9% for biotechnology | VWR | Cat # 0241-5KG |
| Sodium phosphate dibasic | Sigma-Aldrich | Cat # S0876-500G |
| Sodium phosphate monobasic (1Kg) | LPS Solution | Cat # SPM01 |
| Sucrose | JUNSEI | Cat # 31365-0350 |
| Potassium chloride | JUNSEI | Cat # 18190-0350 |
| Potassium chloride 99–100.5% ACS | VWR | Cat # 0395-500 G |
| Tris (Base) | Biosesang | Cat # TR1016-100-00 |
| SDS (sodium dodecyl sulfate) | Biosesang | Cat # SR1010-500-00 |
| Normal goat serum | Jackson ImmunoResearch | Cat # 005-000-121 |
| Normal donkey serum | Jackson ImmunoResearch | Cat # 017-000-121 |
| Potassium phosphate (monobasic) | LPS Solution | Cat # PPM01 |
| Papain | Sigma | Cat # P3375 |
| Fetal bovine serum | Gibco | Cat # 16000044 |
| 10x DPBS | Welgene | Cat # LB 201-02 |
| Triton X-100 | Sigma | Cat # T8787 |
| Tween 20 | Sigma | Cat # P1379 |
| Methanol | Merk | Cat # 106009 |
| Vectashield Plus antifade mounting medium | Vector Laboratories | Cat # H-1900 |
| Neurobasal™ Plus medium | Gibco | Cat # A3582901 |
| B-27™ supplement (50X), serum free | Gibco | Cat # 17504044 |
| Poly-D-lysine hydrobromide | Sigma-Aldrich | Cat # P7886 |
| Natural mouse laminin | Invitrogen | Cat # 23017-015 |
| L-Cysteine | Sigma-Aldrich | Cart # C7352 |
| FITC-conjugated isolectin B4 | Sigma-Aldrich | Cat # L2895 |
| Immobilon®-P PVDF membrane | Millipore | Cat # IPVH00010 |
| mini Trans-Blot filter paper | Bio-Rad | Cat # 1703932 |
| Cell scraper | SPL | Cat # 90020 |
| 2 ml serological pipet | Falcon | Cat # 357507 |
| 5 ml serological pipet | SPL | Cat # 91005 |
| 10 ml serological pipet | SPL | Cat # 91010 |
| 25 ml serological pipet | SPL | Cat # 91025 |
| Cell culture flask 75 | Thermo Fisher | Cat # 156499 |
| Cell culture multi-dish 6 | Thermo Fisher | Cat # 140675 |
| 100 µm cell strainer | SPL | Cat # 93100 |
| 96-well glass-bottom plate | Cellvis | Cat # P96-1.5H-N |

| Reagent/resource | Reference or source | Identifier or catalog number |
|---|---|---|
| 20 mm glass-bottom cell culture dish | Nest | Cat # 801001 |
| Microslide glass | MUTO | Cat # 5116-20 F |
| Cover glass | Mediscope | Cat # HSU-0101192 |
| Falcon 5 mL round-bottom tube | BD | Cat # 352052 |
| **Software** | | |
| pClamp | Molecular Devices | 10.5 |
| Digidata 1550 A | Axon Instruments | 1550 A |
| SnapGene® | SnapGene | 7.2 |
| IMARIS x64 | | 10.7.0.3 |
| Clampex | Molecular Devices | 10.7 |
| R 4.3.1 | R core Team | https://www.r-project.org/ |
| SynBot | Savage et al, 2024 | https://www.protocols.io/view/quantifying-synaptic-colocalizations-with-synbot-m-kxygx9rjog8j/v1 |
| Puncta analyzer | Ippolito and Eroglu, 2010 | https://sites.duke.edu/eroglulab/tools/ |
| GraphPad | Prism | Version 9.2.0 |
| MATLAB | MathWorks | R2021a |
| ZEN | Zeiss | 3.9 |
| EzCalcium | Cantu et al, 2020 | https://matlab.mathworks.com/open/fileexchange/v1?id=109840 |
| Biorender | Biorender | https://app.biorender.com |
| FiJI (ImageJ) | ImageJ | 2.14.0/1.54 f |
| **Other** | | |
| Immobilon®-P PVDF membrane | Millipore | Cat # IPVH00010 |
| mini Trans-Blot filter paper | Bio-Rad | Cat # 1703932 |
| Cell scraper | SPL | Cat # 90020 |
| 2 ml serological pipet | Falcon | Cat # 357507 |
| 5 ml serological pipet | SPL | Cat # 91005 |
| 10 ml serological pipet | SPL | Cat # 91010 |
| 25 ml serological pipet | SPL | Cat # 91025 |
| Cell culture flask 75 | Thermo Fisher | Cat # 156499 |
| Cell culture multi-dish 6 | Thermo Fisher | Cat # 140675 |
| 100 µm cell strainer | SPL | Cat # 93100 |
| 96-well glass-bottom plate | Cellvis | Cat # P96-1.5H-N |
| 20 mm glass-bottom cell culture dish | Nest | Cat # 801001 |
| Microslide glass | MUTO | Cat # 5116-20 F |

| Reagent/resource | Reference or source | Identifier or catalog number |
|---|---|---|
| Cover glass | Mediscope | Cat # HSU-0101192 |
| Falcon 5 mL round-bottom tube | BD | Cat # 352052 |

## Mice

Animal experiments were approved by the Institutional Animal Care and Use Committee of Seoul National University. Male C57BL/6J mice (8–10 weeks of age) were purchased from Daehan Biolink (DBL, Eumsung, Korea). The global *St3gal2,3*$^{-/-}$ mice were generated as previously described (Sturgill et al, 2012). In addition, sensory neuron-specific *St3gal2*-depleted mice were generated by cross-breeding *St3gal2*-floxed mice with *Scn10a-Cre* knock-in mice, which express Cre under the control of the endogenous Na$_v$1.8 promoter. All animals were housed and maintained at 22–24 °C and 55% humidity with a 12-h light/dark cycle in a specific pathogen-free environment. They were given access to food and water ad libitum.

## Neuropathic pain model

Right L5 SNT was performed as previously described (Kim et al, 2007). Briefly, animals were anesthetized with isoflurane in an O$_2$ carrier (induction 2% and maintenance 1.5%), and a small incision exposed the L4 and L5 spinal nerves. The L6 transverse process was partially removed and the L5 spinal nerve was transected. The wound was tightly sutured with surgical skin staples.

## Behavioral testing

Mechanical allodynia tests were performed as previously reported (Lee et al, 2023b). All behavioral tests took place between 10 am and 3 pm, and the experimenter was blind to group assignments throughout the experiment. Mechanical sensitivity of the right hind paw was assessed using a calibrated series of von-Frey hairs (0.02–6 g; Stoelting, Wood Dale, IL, USA) and following an up-down method (Chaplan et al, 1994). Thermal sensitivity was determined by measuring paw withdrawal latencies in response to radiant heat (Hargreaves et al, 1988). Rapid paw withdrawal, licking, and flinching were interpreted as pain responses. Tests were performed after at least three habituations at 24-h intervals. Assessments were made 1 day before surgery for baseline and 1, 3, and 7 days after surgery or injection.

## Immunohistochemistry

Mice were transcardially perfused with 0.1 M phosphate buffer (pH 7.4), followed by 4% paraformaldehyde, and the L5 spinal cord and DRG were removed and post-fixed overnight in the same solution at 4 °C. Spinal cord and DRG samples were transferred to 30% sucrose for at least 48 h and coronally cut into 16-μm-thick sections using a cryostat (CM1860; Leica, Wetzlar, Germany). The spinal cord sections were blocked in a solution containing 5% normal goat serum, 2.5% bovine serum albumin (BSA), and 0.2% Triton X-100 for 1.5 h at room temperature. Then, spinal cord sections were

incubated with rabbit anti-Iba-1 (019-19741, 1:1,000; Wako, Osaka, Japan), mouse anti-glial fibrillary acidic protein (GFAP) (MAB360, 1:1000; Abcam, Cambridge, UK), rabbit anti-GFAP (Z0334, 1:1000; Dako, Santa Clara, CA, USA), mouse anti-VGLUT2 (ab79157, 1:1000; Abcam), rabbit anti-VGLUT2 (ab216463, 1:1000; Abcam), rabbit anti-PSD95 (51-6900, 1:1000; Invitrogen, Waltham, MA, USA), rabbit anti-VGAT (131 006, 1:1000; Synaptic Systems, Goettingen, Germany), mouse anti-VGAT (147 011, 1:000; Synaptic Systems), mouse anti-GT1b (MAB5608, 1:1000; Millipor-eSigma, Burlington, MC, USA), rabbit anti-MAP2 (AB5622, 1:1000; MilliporeSigma), or mouse anti-CGRP (AB81887, 1:1000; Abcam). After being rinsed 5 times with 0.1 M PBS, the samples were incubated with FITC-, CY3-, and Alexa Fluor 647-conjugated secondary antibodies (1:200; Jackson ImmunoResearch Laboratories, West Grove, PA, USA), and Alexa Fluor 405-conjugated antibody (A-31553, 1:200; Invitrogen) for 1.5 h at room temperature. The samples were then mounted on glass slides with Vectashield mounting medium (Vector Laboratories, Burlingame, CA, USA).

## Image acquisition and analysis

Images were captured using LSM 980 and LSM800 confocal microscopes (Carl Zeiss, Oberkochen, Germany).

## Quantification of spinal synapses

To quantify spinal synapses, 10 focal planes from three regions of both the ipsilateral and contralateral sides were captured at 63x magnification, with a 1 μm interval (10 sections/Z-stack, image size, 101.41 μm × 101.41 μm). Then, images were analyzed using a puncta analyzer and SynBot using ImageJ software, as previously described (Ippolito and Eroglu, 2010; Savage et al, 2024). Briefly, the spinal cord images were immunostained with pre- and post-synaptic markers. The images were separated into red and green channels, underwent background subtraction with a rolling ball radius of 50 pixels, and were thresholded to quantify co-localized puncta by identifying those that closely aligned across the two channels.

## 3D analysis

For 3D reconstruction of microglia and astrocytes, we obtained Z-stack images (9 μm depth, 130 nm steps, 70 slides) of spinal dorsal horns. These images were acquired using an LSM 980 Airy scan microscope (2924 × 2924 pixels, 16-bit depth, 103.18 μm pixel size). The raw image files (.czi) were converted and analyzed using IMARIS (Version 9.8.0, Oxford Instruments, Abingdon, UK). A surface for each fluorescence signal was generated for synaptic puncta (surface gain size, 0.150 μm; diameter of largest sphere, 0.571 μm; growth region estimated diameter, 1 μm). Normal synapses and phagocytosed synaptic puncta were classified based on the shortest distance and overlap volume between EGFP and mCherry puncta. Similarly, the analysis of synaptic phagocytosis and attachment was conducted using measurements of the shortest distance and overlap volume between synaptic puncta and glial structures.

## Drug administration

To inhibit the synthesis of GT1b, 5 μl of diluted PDMP (Cat# 1756, Matreya LLC, PA, USA) was administered. Intrathecal injection

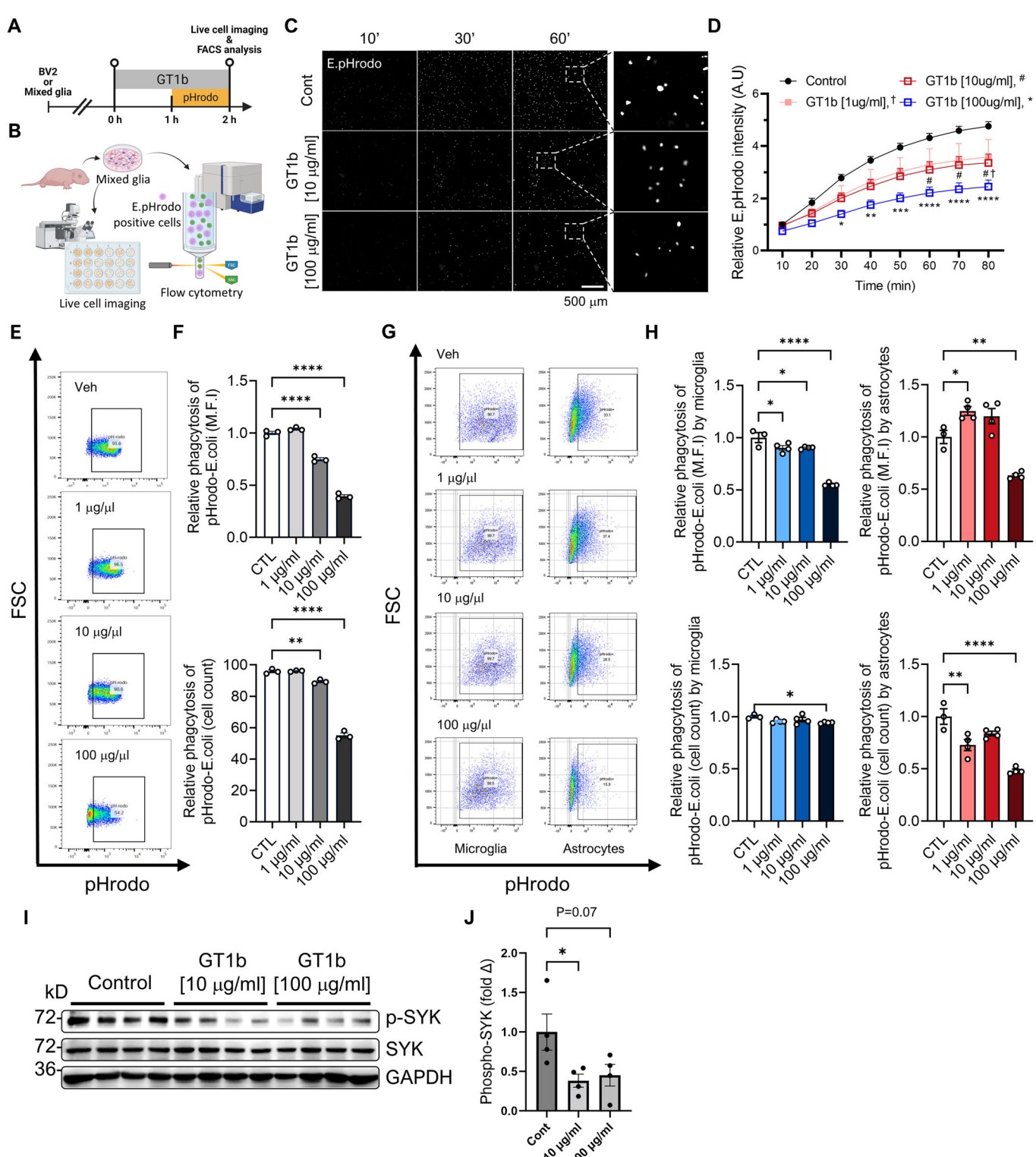

was performed on mice anesthetized with 1.5–2% isoflurane in an $O_2$ carrier. Each drug was administered using a 10-μl Hamilton syringe (Hamilton Co., Reno, NV, USA) with a 30-G needle, as previously described (Lee et al, 2023a). The success of each intrathecal injection was assessed by monitoring a slight tail-flick when the needle penetrated the subarachnoid space.

## Synapse-specific pH-reporter expression

To investigate the interaction between glia and synapses, we used a synapse-specific pH-reporter system (ExPre, pAAV2retro::CaMKIIa-Synaptophysin-mCherry-eGFP; ExPost, pAAV9-CW3SL::hSyn-PSD95-mCherry-eGFP; InhiPre, pAAV9::GAD67-synaptophysin-mCherry-eGFP;

◀ **Figure 5. GT1b inhibits glial phagocytosis via SYK dephosphorylation.**

(A) Experimental timeline for phagocytosis assays using BV2 and primary mixed glial cells after GT1b stimulation. (B) Overview of live cell imaging and flow cytometry for the phagocytosis assay. (C) Representative image of the glial phagocytosis of *E. coli*-conjugated pHrodo bioparticles after GT1b treatment. Scale bar = 500 µm. (D) Quantification of glial phagocytic activity. *P* value (30 min); Control vs. GT1b[100 µg/ml] = 0.0136. *P* value (40 min); Control vs. GT1b[100 µg/ml] = 0.0017. *P* value (50 min); Control vs. GT1b[100 µg/ml] = 0.0003. *P* value (60 min); Control vs. GT1b[10 µg/ml] = 0.0333, Control vs. GT1b[100 µg/ml] <0.0001. *P* value (70 min); Control vs. GT1b[10 µg/ml] = 0.02, Control vs. GT1b[100 µg/ml] <0.0001. *P* value (80 min); Control vs. GT1b[1 µg/ml] = 0.0414, Control vs. GT1b[10 µg/ml] =0.0117, Control vs. GT1b[100 µg/ml] <0.0001. *n*; Control = 3, GT1b[1 µg/ml] = 4, GT1b[10 µg/ml] = 4, GT1b[100 µg/ml] = 4. (E) Gating strategy for BV2 cells and (G) mixed glia, with corresponding phagocytic activity levels depicted in (F) and (H), respectively. *P* value (F: M.F.I); CTL vs. 1 µg/ml = 0.2108, CTL vs. 10 µg/ml <0.0001, CTL vs. 100 µg/ml <0.0001. *P* value (F: cell count); CTL vs. 1 µg/ml = 0.9947, CTL vs. 10 µg/ml = 0.0041, CTL vs. 100 µg/ml <0.0001. *n* (F); CTL = 3, 1 µg/ml = 3, 10 µg/ml = 3. 100 µg/ml = 3. *P* value (H: microglia_M.F.I); CTL vs. 1 µg/ml = 0.0342, CTL vs. 10 µg/ml = 0.0341, CTL vs. 100 µg/ml <0.0001. *P* value (H: microglia_cell count); CTL vs. 1 µg/ml = 0.1046, CTL vs. 10 µg/ml = 0.3841, CTL vs. 100 µg/ml = 0.0349. *P* value (H: astrocyte_M.F.I); CTL vs. 1 µg/ml = 0.0198, CTL vs. 10 µg/ml = 0.0611, CTL vs. 100 µg/ml =0.0015. *P* value (H: astrocyte_cell count); CTL vs. 1 µg/ml = 0.003, CTL vs. 10 µg/ml = 0.0598, CTL vs. 100 µg/ml <0.0001. *n* (H); CTL = 3, 1 µg/ml = 4, 10 µg/ml = 4. 100 µg/ml = 4. (I) Western blot representation and (J) quantification of phosphorylated and total SYK levels in primary mixed glia. *P* value; Control vs. 10 µg/ml = 0.0432, CTL vs. 100 µg/ml = 0.07. *n*; Cont = 4, 10 µg/ml = 4. 100 µg/ml = 4. Data are presented as the mean ± SEM; *P < 0.05, **P < 0.01, ***P < 0.001, ****P < 0.0001; one-way ANOVA with Dunnett's multiple comparison test and two-way ANOVA with Dunnett's multiple comparison test. Source data are available online for this figure.

InhiPost, pAAV9-CW3SL::hSyn-gephyrin-mCherry-eGFP) (Lee et al, 2021a). To express the pH-reporter system in excitatory pre-synapses projecting to the LI–II region, we replaced the human synapsin (hSyn) promoter in the ExPre plasmid with CaMKIIa. The AAV2 retrovirus was produced by the Korea Institute of Science and Technology Virus. We used a syringe pump (Stoelting Co.) and a 10-µl Hamilton syringe (Hamilton Co.) to deliver 0.5 µl of AAV virus to the ipsilateral side of the spinal dorsal horn, as previously described (150 µm below the surface) (Kohro et al, 2015). Briefly, the animals were anesthetized with isoflurane in an $O_2$ carrier (induction 2% and maintenance 1.5%). A small incision exposed the paraspinal muscles, and then an incision around the T13 and L1 vertebrae created a small window for accessing the spinal cord. The dura mater and arachnoid membrane were incised with a 30 G needle, allowing the insertion of the syringe. The virus was administered at a constant rate of 200 nl/min, and the needle was left in place for 10 min after each injection to minimize the potential upward flow of the viral solution.

## Electrophysiology

L4–L6 spinal cord slices (300 µm) were prepared (Zhu et al, 2019). Briefly, the mice were anesthetized with isoflurane, and the spinal cords were rapidly extracted. The meninges, pia-arachnoid membrane, and attached spinal nerve roots were carefully removed. Each spinal cord was then mounted on an agarose block, and coronal slices were obtained using a vibratome (VT1000 S, Leica). These slices were placed in ice-cold, oxygenated (95% $O_2$ and 5% $CO_2$) dissection buffer with a low-$Ca^{2+}$/high-$Mg^{2+}$ concentration (5 mM KCl, 1.23 mM $NaH_2PO_4$, 26 mM $NaHCO_3$, 10 mM dextrose, 0.5 mM $CaCl_2$, 10 mM $MgCl_2$, 212.7 mM sucrose). Afterward, the slices were moved to a holding chamber in an incubator at 28–30°C containing oxygenated (95% $O_2$ and 5% $CO_2$) artificial cerebrospinal fluid (ACSF: 124 mM NaCl, 5 mM KCl, 1.23 mM $NaH_2PO_4$, 26 mM $NaHCO_3$, 10 mM dextrose, 2.5 mM $CaCl_2$, and 1.5 mM $MgCl_2$), where they were allowed to rest for at least 1 h before recording. Once they had sufficiently recovered, the slices were transferred to a recording chamber oxygenated with 95% $O_2$ and 5% $CO_2$ at a flow rate of 2 ml/min, where they were continuously perfused with ACSF. The slices were allowed to equilibrate for 5 min before the start of the recordings, and all experiments were conducted at a temperature range of 30–32 °C.

The recordings were acquired using a Multiclamp 700B amplifier (Molecular Devices, Sunnyvale, CA, USA) under visual control using an upright microscope (BX51WI; Olympus, Tokyo, Japan) with differential interference contrast illumination. Patch pipettes with resistance ranging from 4–6 MΩ were filled with a solution containing 135 mM K-gluconate, 8 mM NaCl, 10 mM HEPES, 2 mM ATP-Na, and 0.2 mM GTP-Na to record the sEPSCs in voltage-clamp mode (pH 7.4 and 280–290 mOsm). Only cells with an access resistance less than 20 MΩ and an input resistance greater than 100 MΩ were analyzed. Data were acquired and analyzed using pClamp 10.5 (Molecular Devices). Signals were filtered at 2 kHz and digitized at 10 kHz using Digidata 1550 A (Axon Instruments, Union City, CA, USA).

## In vitro DRG neuron and microglial culture

Primary DRG neurons were isolated from a mouse embryo at E15, as previously described (Griso and Puccio, 2020). Briefly, DRGs were harvested from an E15 mouse embryo in ice-cold HBSS buffer with 2% D-glucose. Subsequent steps included enzymatic dissociation with papain and collagenase complemented by mechanical dissociation via pipetting. The quantified DRG neurons were deposited into 96-well glass bottom plates (P96-1.5H-N, Cellvis, Mountain View, CA, USA) pre-coated with poly-D-lysine and laminin. The DRG neurons were maintained with Neurobasal Plus medium (A3582901, Gibco, Waltham, MC, USA) supplemented with 0.1% fetal bovine serum (FBS), 1× B-27, and 2 mM L-glutamine. Replacement of 30% of the medium was performed every 5 days. Primary mixed glial cultures were prepared from 1–2-day-old mice, as previously described (Lee et al, 2000). In brief, brain glial cells were cultured in high-glucose Dulbecco's modified Eagle medium (DMEM), supplemented with 10% FBS, 10 mM HEPES, 2 mM L-glutamine, 1× penicillin/streptomycin, and 1× nonessential amino acid mixture at 37 °C in a 5% $CO_2$ incubator. The medium was renewed every 5 days. After 15 days, primary mixed glia were harvested with 0.25% trypsin-EDTA and seeded into six-well plates. BV2 cells, a murine microglial cell line, were maintained at 37 °C and 5% $CO_2$ in a humidified incubator. The growth medium for the BV2 cells consisted of high-glucose DMEM supplemented with 10% FBS, 10 mM HEPES, 2 mM L-glutamine, 1× penicillin/streptomycin, and 1× nonessential amino acid

mixture. The medium was renewed every 3 days, and the BV2 cells were subcultured when they reached 80% confluence.

## In vitro live imaging

### Ca$^{2+}$ recording

To assess spontaneous pre-synaptic calcium activity, cultured DRG neurons expressing synaptophysin-conjugated GCaMP7f were subjected to examination using a confocal microscope (LSM800, Carl Zeiss) in a controlled environment at 37 °C and 5% $CO_2$ within a humidified chamber. DRG neurons were first transitioned to Tyrode's buffer (T1788, MilliporeSigma). A comprehensive time-lapse imaging session spanning 10 min was executed. The analysis of calcium activity was facilitated using the EzCalcium open-source MATLAB toolbox (Cantu et al, 2020). In this process, specific regions of interest (ROIs) within the pre-synaptic compartments were designated, and relative fluorescence changes were quantified. The activity in each ROI of the acquired images was collected and categorized based on the presence or absence of interactions with GT1b.

## GT1b visualization

To visualize GT1b, CF647 was conjugated to GT1b antibody (MAB5608, 1:1000; MilliporeSigma) using a Mix-n-Stain CF dye antibody labeling kit (MX647S100, MilliporeSigma). DRG neurons were subsequently incubated with the CF647-labeled GT1b antibody for 1 h. After incubation, the DRG neurons were gently washed with Tyrode's buffer to remove any unbound antibodies.

## Phagocytosis assay

pHrodo™ Red *E. coli* BioParticles™ Conjugate (P35361, Invitrogen) was used for the phagocytosis assay. BV2 cells were subcultured in six-well plates and used once they reached ~80% confluence. Mixed glial cells were seeded in six-well plates at a concentration of $1 \times 10^6$ cells per well. Following a 4-h serum-free incubation, GT1b was administered at various concentrations (1 µg/ml, 10 µg/ml, and 100 µg/ml). Subsequently, the pHrodo™ Red *E. coli* BioParticles™ conjugate was added and allowed to incubate for 1 h.

## Live cell imaging

After being thoroughly washed to remove any remaining bioparticles, mixed glial cells were examined using a multimode automated system for live cell imaging (Image ExFluorer, LCI). The examination was conducted in a controlled environment of 37 °C and 5% $CO_2$ with humidity. A comprehensive time-lapse imaging session was executed over a span of 80 min to assess phagocytic activity. The analysis was facilitated by the toolbox included with the system. Six ROIs were designated within each compartment of a six-well plate, with the average value of each well counted as $n = 1$. Relative changes in fluorescence were quantified to evaluate phagocytic activity.

## Flow cytometry analysis

After being thoroughly washed to remove any remaining bioparticles, BV2 cells were detached using a cell scraper, and

mixed glial cells were trypsinized with 0.25% trypsin-EDTA (25200056, Gibco). The cells were subjected to staining for flow cytometry. Briefly, an Fc blocker was used to minimize potential non-specific antibody staining through a 1-h incubation. Subsequently, anti-CD11b-bv421 (101235, BioLegend, San Diego, CA, USA) and anti-ACSA-2-APC (130-102-315, Miltenyi Biotec, Lund, Sweden) were used to differentiate microglia and astrocytes. Stained cells were then filtered through a 70-µm strainer and analyzed using FACSverse and FACSaria.

## Western blotting

Primary mixed glial cells were subjected to an administration of GT1b as described above. After 1 h of GT1b stimulation, the mixed glial cells were lysed in RIPA buffer containing 50 nM Tris-HCL (pH 7.5), 150 nM NaCl, 1% NP-40, 0.5% deoxycholic acid, 0.1% SDS, 1 mM PMSF, and phosphatase inhibitor mixture set IV (p5726, Sigma-Aldrich, MO, USA). Then, the lysates were centrifuged at 13,000 rpm for 15 min at 4 °C, and the supernatants were collected. The concentration of proteins was quantified using Pierce BCA protein assay kits (23225, Thermo Fisher, MA, USA), and 20 µg of protein was mixed with 5x SDS-PAGE sample buffer. Those samples were separated using 10% SDS-PAGE and transferred to a methanol-activated PVDF membrane (IPVH00010, Millipore, MA, USA). The membrane was blocked with 5% BSA in Tris-buffered saline containing Tween 20 (TBST) and incubated overnight with primary antibodies in 2.5% BSA/TBST at 4 °C with gentle agitation. The membrane was developed by enhanced chemiluminescence reagents with HRP-conjugated secondary antibodies, and immunoreactivity was measured with a Fusion FX6.0 (Viver Lourmat, Sursee, Switzerland). For normalization, the membrane was re-proved with mouse GADPH (Sigma-Aldrich) antibody. Data were analyzed with ImageJ.

## Illustrations

Illustrations were generated with Biorender (https://www.biorender.com/).

## Statistics

Data were analyzed using Student's *t* test for comparisons between two groups. One- and two-way analysis of variance (ANOVA) with Tukey's and Dunnett's multiple comparison tests or Bonferroni's post-hoc test were used for statistical analysis of multiple comparisons, as specified in the figure legends. All data are presented as the mean ± standard error of the mean (SEM), and differences were considered significant at a $P$ value < 0.05.

# Data availability

This paper does not report any original code or deposited data.

The source data of this paper are collected in the following database record: biostudies:S-SCDT-10_1038-S44319-025-00452-2.

## Peer review information

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

## Acknowledgements

We extend our appreciation to all members of the Lee laboratory, particularly Won-Suck Chung and Sungho Chang, for their feedback and support throughout the study. We thank Seog-Bae Oh and Se-Young Choi for scientific discussion that contributed to this manuscript. We thank the members of the Dental Research Institute (DRI) facilities center for technical assistance and Cagla Eroglu for providing tools for synapse counting. This study was supported by the National Research Foundation of Korea (2022R1A2C1091994 and RS-2024-00353822).

## Author contributions

**Jaesung Lee**: Conceptualization; Resources; Data curation; Software; Formal analysis; Supervision; Funding acquisition; Validation; Investigation; Visualization; Methodology; Writing—original draft; Project administration; Writing—review and editing; first author. **Kyungchul Noh**: Data curation. **Subeen Lee**: Data curation. **Kwang Hwan Kim**: Data curation. **Seohyun Chung**: Data curation. **Hyoungsub Lim**: Resources. **Minkyu Hwang**: Data curation. **Joon-Hyuk Lee**: Resources. **Won-Suk Chung**: Resources; Supervision. **Sunghoe Chang**: Resources; Supervision; Methodology. **Sung Joong Lee**: Conceptualization; Resources; Supervision; Funding acquisition; Writing—original draft; Project administration; Writing—review and editing.

Source data underlying figure panels in this paper may have individual authorship assigned. Where available, figure panel/source data authorship is listed in the following database record: biostudies:S-SCDT-10_1038-S44319-025-00452-2.

## Disclosure and competing interests statement

The authors declare no competing interests.

# Expanded View Figures

**Figure EV1.  AAV-based pH-reporter expression in each type of synapse and its attachment to spinal glia.**

(**A**) Targeted expression of the synapse type–specific pH-reporter system, which was introduced via AAV vectors. Representative immunostained images demonstrate the colocalization of the pH-reporter with each synaptic marker. Scale bar = 10 µm. (**B**) Comparative analysis of ExPre attachment to each glial cell type. *n*; Sham = 8, 3 d = 15, 7 d = 7. (**C**) Comparative analysis of ExPost attachment to each glial cell type. *P* value (Sham); Microglia vs. Astrocyte <0.0001. *P* value (3 d); Microglia vs. Astrocyte <0.0001. *n*; Sham = 9, 3 d = 9, 7 d = 6. (**D**) Comparative analysis of InhiPre attachment to each glial cell type. *P* value (Sham); Microglia vs. Astrocyte = 0.0272. *n*; Sham = 6, 3 d = 7, 7 d = 6. (**E**) Comparative analysis of InhiPost attachment to each glial cell type. *P* value (Sham); Microglia vs. Astrocyte = 0.0004. *P* value (3 d); Microglia vs. Astrocyte = 0.0418. P value (7 d); Microglia vs. Astrocyte <0.0001. *n*; Sham = 8, 3 d = 7, 7 d = 8. Data are represented as the mean ± SEM; *$P < 0.05$, ***$P < 0.001$, ****$P < 0.0001$; two-way ANOVA with Tukey's multiple comparison test.

                                                    

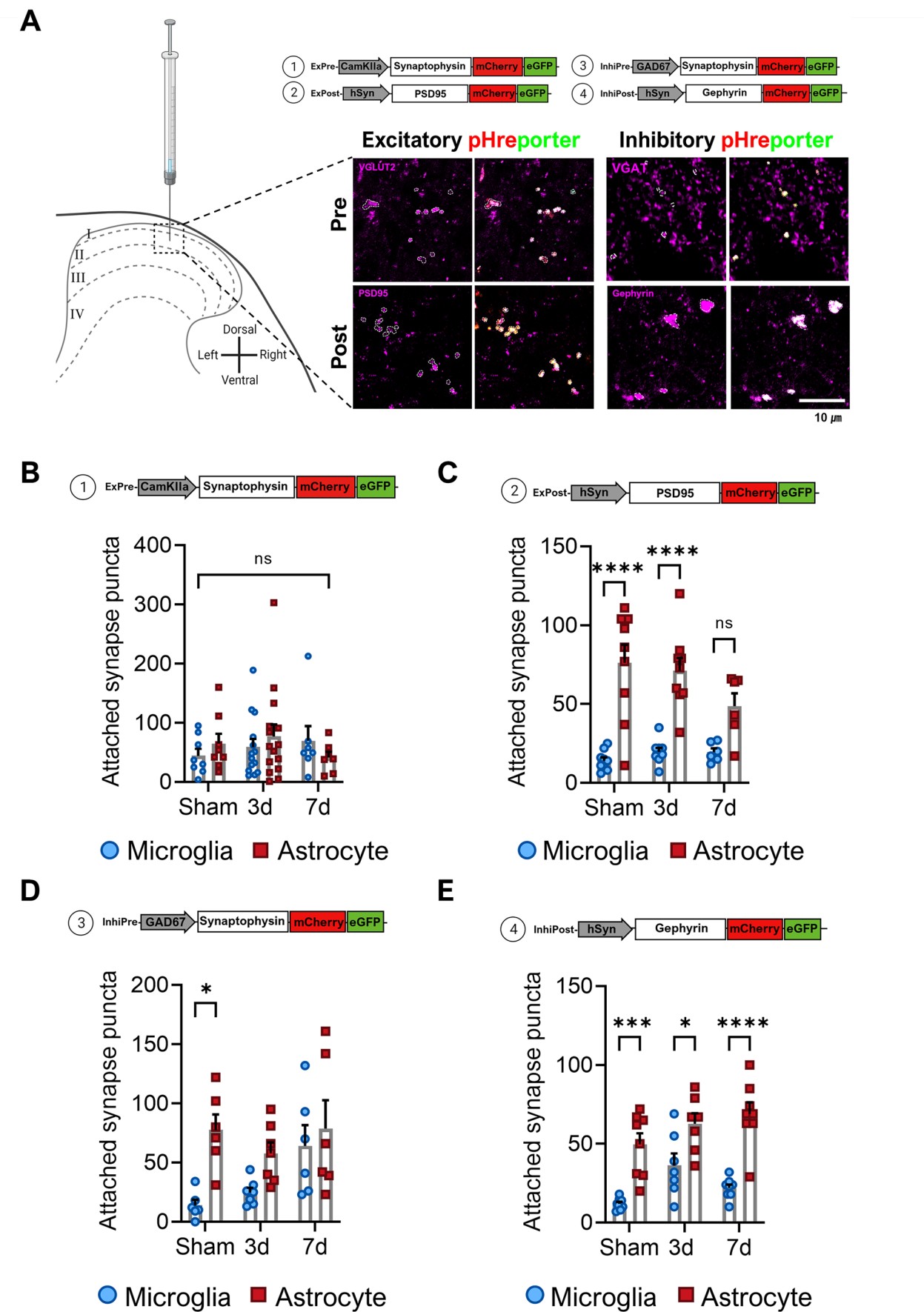

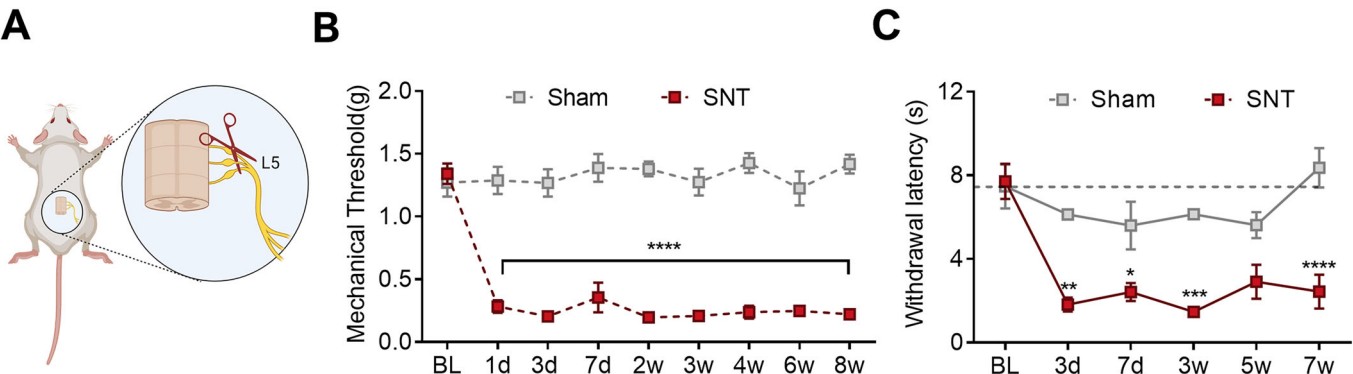

**Figure EV2. L5 spinal nerve transection (L5 SNT) model of neuropathic pain.**

(A) Schematic of the L5 SNT model used to induce neuropathic pain in mice. (B) Assessment of mechanical allodynia and (C) thermal hyperalgesia. $P$ value (mechanical allodynia); Sham vs. SNT < 0.0001. $n$ (mechanical allodynia); Sham = 9, SNT = 16. $P$ value (thermal hyperalgesia); 3 d:Sham vs. 3 d:SNT = 0.0016, 7 d:Sham vs. 7 d:SNT = 0.0256, 3w:Sham vs. 3w:SNT = 0.0007, 5w:Sham vs. 5w:SNT = 0.0779, 7w:Sham vs. 7w:SNT < 0.0001. Sham = 3, SNT = 3. Data are represented as the mean ± SEM; *$P$ < 0.05, ***$P$ < 0.001, ****$P$ < 0.0001; two-way ANOVA with Bonferroni's multiple comparison test.

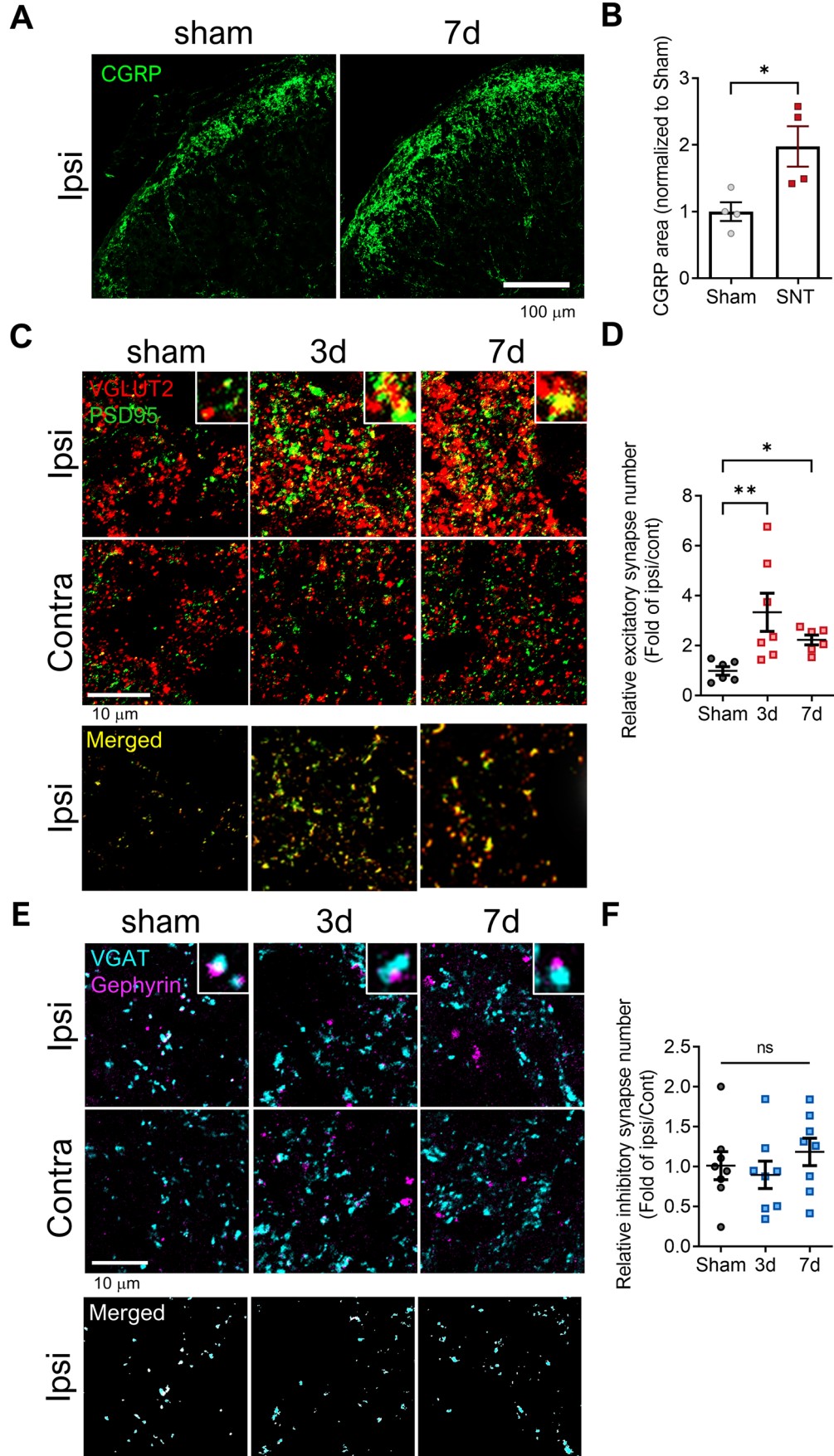

**Figure EV3. Expansion of excitatory synapses in the superficial layer (LI–II) of the spinal cord.**

(A, B) Immunofluorescence visualization of calcitonin gene-related peptide (CGRP) in the dorsal horn, depicting and quantifying the expansion of afferent terminals after nerve injury. Scale bar = 100 μm. *P* value; Sham vs. SNT = 0.0267. *n*; Sham = 4, SNT = 4. (C) High-resolution confocal images of spinal cord sections stained for VGLUT2 (red) and PSD95 (green) to label excitatory synapses. Insets show a magnified view of co-localized VGLUT2 + PSD95+ puncta. (D) Quantification of co-localized excitatory synaptic puncta following SNT. *P* value; Sham vs. 3 d = 0.0036, Sham vs. 7 d = 0.0153. *n*; Sham = 6, 3 d = 7, 7 d = 6. Scale bar = 10 μm. (E) High-resolution confocal images of spinal cord sections stained for VGAT (cyan) and gephyrin (magenta) to label inhibitory synapses. Insets show magnified view of co-localized VGAT+ gephyrin+ puncta. (F) Quantification of co-localized inhibitory synaptic puncta after SNT. Scale bar = 10 μm. *n*; Sham = 8, 3 d = 8, 7 d = 8. Data are presented as the mean ± SEM; *$P < 0.05$, **$P < 0.01$; one-way ANOVA with Dunnett's multiple comparison test and Student's *t* test.

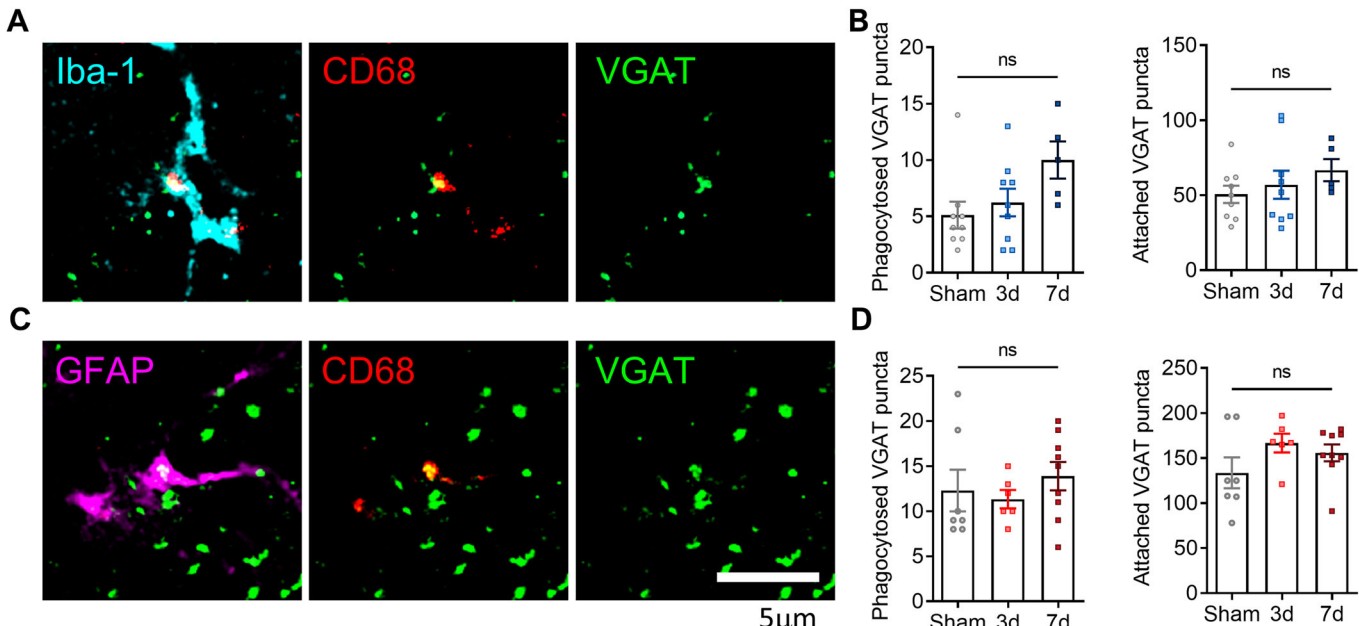

**Figure EV4. Glial phagocytosis of VGAT-positive InhiPre after peripheral nerve injury.**

(A) Representative immunostained spinal cord section showing VGAT, CD68 and Iba-1, highlighting microglial phagocytosis of VGAT-positive InhiPre. (B) Microglial phagocytosis and surface attachment of VGAT-positive InhiPre. *n*; Sham = 9, 3 d = 9, 7 d = 5. (C) Representative immunostained spinal cord section showing VGAT, CD68 and GFAP, illustrating astrocytic phagocytosis of VGAT-positive InhiPre. (D) Astrocytic phagocytosis and surface attachment of VGAT-positive InhiPre. *n*; Sham = 7, 3 d = 6, 7 d = 9. Scale bar = 5 μm. Data are represented as the mean ± SEM; one-way ANOVA with Dunnett's multiple comparison test.

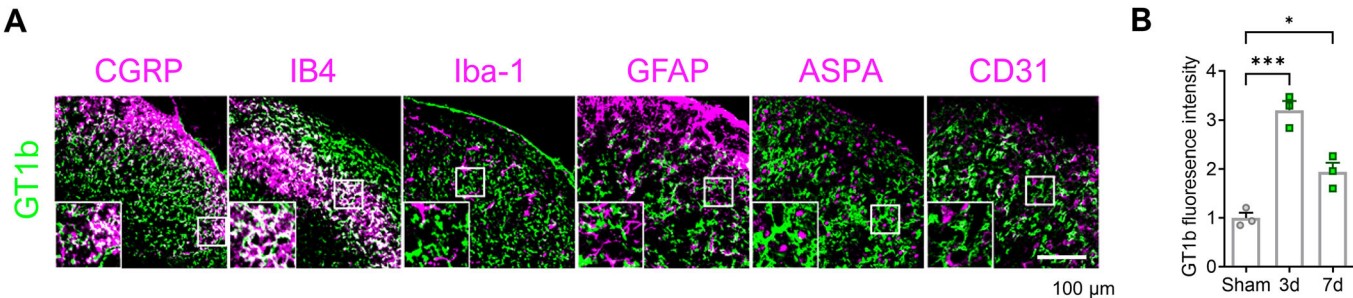

**Figure EV5.  Localization of GT1b in spinal afferent terminals and its accumulation following peripheral nerve injury.**

(A) Immunostained spinal cord sections demonstrating GT1b localization with markers for various cell types: CGRP (calcitonin gene-related peptide) for peptidergic afferents, IB4 (isolectin B4) for non-peptidergic afferents, Iba-1 for microglia, GFAP for astrocytes, ASAP (aspartoacylase) for oligodendrocytes, and CD31 for endothelia. (B) GT1b fluorescence intensity across post-injury time points in the dorsal horn of the spinal cord. $P$ value; Sham vs. 3 d = 0.0002, Sham vs. 7 d = 0.0128. $n$; Sham = 3, 3 d = 3, 7 d = 3. Scale bar = 100 μm. Data are represented as the mean ± SEM; *$P < 0.05$, ***$P < 0.001$; one-way ANOVA with Dunnett's multiple comparison test.

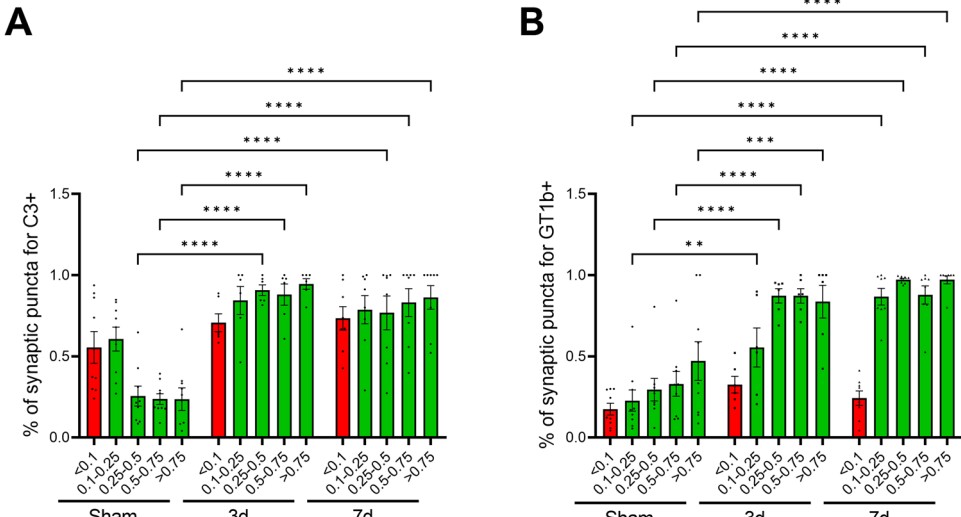

**Figure EV6.   EGFP intensity-dependent ratios of C3- and GT1b-positive synaptic puncta.**

Five categories of ExPre synaptic puncta were defined based on the intensity of EGFP. mCherry puncta with less than 10% of EGFP were classified as phagocytosed puncta, and the remaining puncta were divided into four groups (10–25%, 25–50%, 50–75%, and more than 75% EGFP co-localized with mCherry puncta). (**A**) Relative ratio of C3-positive puncta in each synapse category. *P* value (0.25-0.5); Sham vs. 3 d < 0.0001, Sham vs. 7 d < 0.0001. *P* value (0.5-0.75); Sham vs. 3 d < 0.0001, Sham vs. 7 d < 0.0001. *P* value (>0.75); Sham vs. 3 d < 0.0001, Sham vs. 7 d < 0.0001. (**B**) Relative ratio of GT1b-positive puncta in each synapse category. *P* value (0.1-0.25); Sham vs. 3 d = 0.0029, Sham vs. 7 d < 0.0001. *P* value (0.25-0.5); Sham vs. 3 d < 0.0001, Sham vs. 7 d < 0.0001. *P* value (0.5-0.75); Sham vs. 3 d < 0.0001, Sham vs. 7 d < 0.0001. *P* value (>0.75); Sham vs. 3 d = 0.0008, Sham vs. 7 d < 0.0001. *n*; Sham = 9, 3 d = 6, 7 d = 8. Data are presented as the mean ± SEM; **P < 0.01, ***P < 0.001, ****P < 0.0001; two-way ANOVA with Dunnett's multiple comparison test.

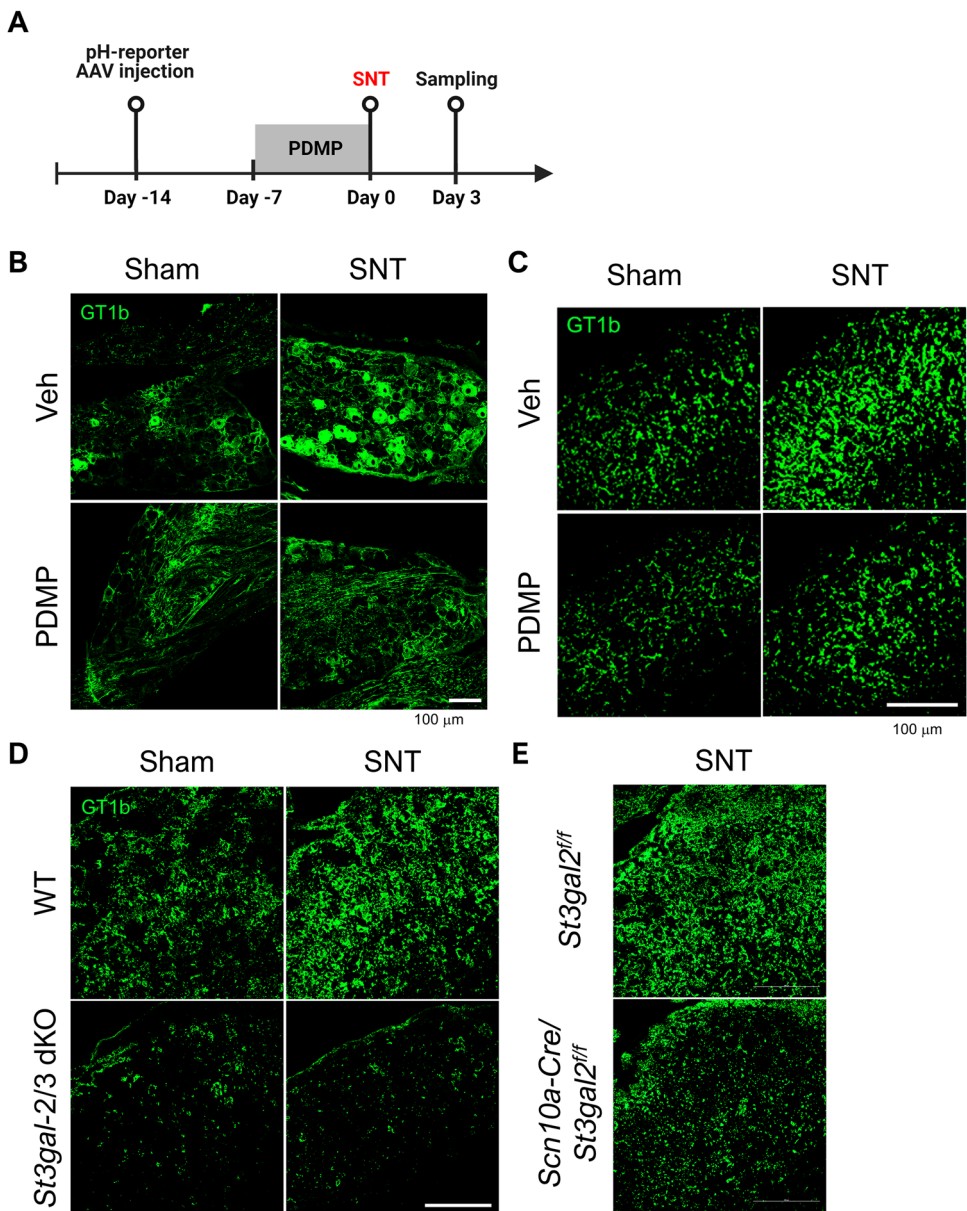

**Figure EV7. Pharmacological and genetic suppression of GT1b synthesis.**

(A) Experimental timeline for in vivo inhibition of GT1b synthesis using PDMP. (B) Representative immunostaining for GT1b in the DRG and (C) spinal cord following nerve injury. (D) Immunostaining of spinal cords from *St3gal2/3* knockout and (E) *Scn10a-Cre/St3gal2^{f/f}* mice with GT1b. Scale bar = 100 μm.

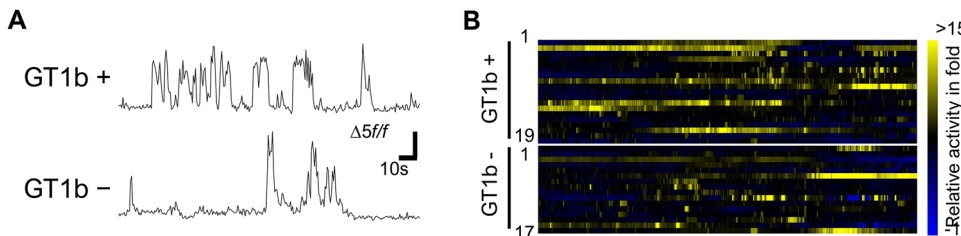

**Figure EV8. Representative trace and heatmap of sCaTs in cultured DRG neurons.**

(**A**) Representative trace and (**B**) heatmap of $Ca^{2+}$ activity.

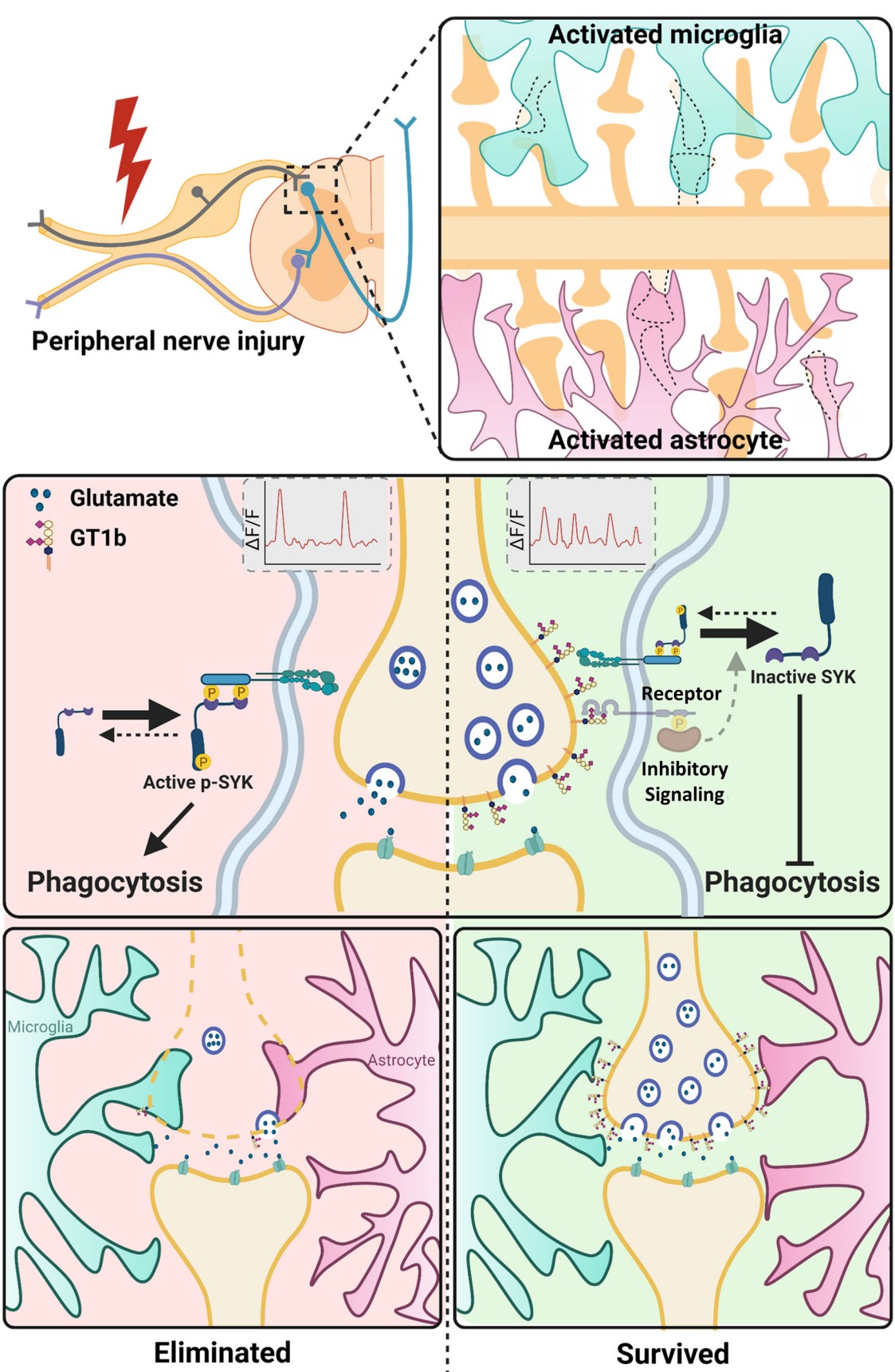

◀  **Figure EV9.  Graphical summary.**

This diagram illustrates the proposed mechanism whereby GT1b acts as a "don't eat me" signal to inhibit glial phagocytosis, preventing the elimination of synapses during spinal synapse remodeling after nerve injury. It shows activated microglia and astrocytes in the vicinity of a synapse and how the presence of GT1b on active synapses can protect them from glial phagocytosis.

