## [Peer Review File · EMBO Reports]

Ganglioside GT1b prevents selective spinal synapse removal following peripheral nerve injury

Jaesung Lee, Kyungchul Noh, Subeen Lee, Kwang Kim, Seohyun Chung, Hyoungsub Lim, Minkyu Hwang, Joon-Hyuk Lee, Won-Suk Chung, Sunghoe Chang, and Sung Joong Lee

Corresponding author(s): Sung Joong Lee (sjlee87@snu.ac.kr) , Sunghoe Chang (sunghoe@snu.ac.kr)

Review Timeline:

Submission Date:	12th Sep 24
Editorial Decision:	16th Dec 24
Revision Received:	11th Feb 25
Editorial Decision:	21st Mar 25
Revision Received:	28th Mar 25
Additional Editor's Correspondence to Author:	3rd Apr 25
Author's Response to Editor's Correspondence:	5th Apr 25
Accepted:	7th Apr 25

Editor: *Esther Schnapp*

Transaction Report:

Dear Prof. Lee,

Thank you for the submission of your manuscript to EMBO reports. We have now received the full set of referee reports as well as referee cross-comments that are all pasted below.

As you will see, the referees acknowledge that the findings are potentially interesting. However, they also point out that the data are over-interpreted and that the ms text needs to be substantially re-written. Referee 1 also agrees with referee 3 that the use of VGAT to detect all inhibitory synapses would clearly strengthen the study. I would therefore like to encourage you to perform these experiments. Please let me know if you have any questions or comments and we can discuss this further, also in a video chat, if you like. All other referee concerns will also need to be addressed.

I would thus like to invite you to revise your manuscript with the understanding that the referee concerns must be fully addressed and their suggestions taken on board. Please address all referee concerns in a complete point-by-point response. Acceptance of the manuscript will depend on a positive outcome of a second round of review. It is EMBO reports policy to allow a single round of major revision only and acceptance or rejection of the manuscript will therefore depend on the completeness of your responses included in the next, final version of the manuscript.

We realize that it is difficult to revise to a specific deadline. In the interest of protecting the conceptual advance provided by the work, we recommend a revision within 3 months (18th Mar 2025). Please discuss the revision progress ahead of this time with the editor if you require more time to complete the revisions.

You can either publish the study as a short report or as a full article. For short reports, the revised manuscript should not exceed 29,000 characters (including spaces but excluding materials & methods and references) and 5 main plus 5 expanded view figures. The results and discussion sections must further be combined, which will help to shorten the manuscript text by eliminating some redundancy that is inevitable when discussing the same experiments twice. For a normal article there are no length limitations, but it should have more than 5 main figures and the results and discussion sections must be separate. In both cases, the entire materials and methods must be included in the main manuscript file.

- 1) A data availability section providing access to data deposited in public databases is missing. If you have not deposited any data, please add a sentence to the data availability section that explains that.
- 2) Your manuscript contains statistics and error bars based on $n=2$. Please use scatter blots in these cases. No statistics should be calculated if $n=2$.

3) We replaced Supplementary Information with Expanded View (EV) Figures and Tables that are collapsible/expandable online. A maximum of 5 EV Figures can be typeset. EV Figures should be cited as 'Figure EV1, Figure EV2' etc... in the text and their respective legends should be included in the main text after the legends of regular figures.

5) a complete author checklist, which you can download from our author guidelines . Please insert information in the checklist that is also reflected in the manuscript. The completed author checklist will also be part of the RPF.

6) Please note that all corresponding authors are required to supply an ORCID ID for their name upon submission of a revised manuscript (). Please find instructions on how to link your ORCID ID to your account in our manuscript tracking system in our Author guidelines

7) Before submitting your revision, primary datasets produced in this study need to be deposited in an appropriate public database (see <https://www.embopress.org/page/journal/14693178/authorguide#datadeposition>). Please remember to provide a reviewer password if the datasets are not yet public. The accession numbers and database should be listed in a formal "Data Availability" section placed after Materials & Method (see also <https://www.embopress.org/page/journal/14693178/authorguide#datadeposition>). Please note that the Data Availability Section is restricted to new primary data that are part of this study. * Note - All links should resolve to a page where the data can be accessed. *
If your study has not produced novel datasets, please mention this fact in the Data Availability Section.

- the name of the statistical test used to generate error bars and P values,
- the number (n) of independent experiments (please specify technical or biological replicates) underlying each data point,
- the nature of the bars and error bars (s.d., s.e.m.),
- If the data are obtained from n {less than or equal to} 2, use scatter blots showing the individual data points.

12) All Materials and Methods need to be described in the main text using our 'Structured Methods' format, which is required for all research articles. According to this format, the Methods section includes a separate Reagents and Tools Table file (listing key reagents, experimental models, software and relevant equipment and including their sources and relevant identifiers) and a Methods and Protocols section describing the methods using a step-by-step protocol format. The aim is to facilitate adoption of the methodologies across labs. More information on how to adhere to this format as well as a downloadable template (.docx) for the Reagents and Tools Table can be found in our author guidelines:
<https://www.embopress.org/page/journal/14693178/authorguide#structuredmethods>.

An example of a Method paper with Structured Methods can be found here: <https://www.embopress.org/doi/full/10.1038/s44320-024-00037-6#sec-4>

I look forward to seeing a revised form of your manuscript when it is ready.

Yours sincerely,

Referee #1:

The manuscript by Lee et al. investigates the role of ganglioside GT1b in glia-dependent synapse elimination after spinal cord injury. The authors demonstrate that GT1b inhibits the uptake of excitatory pre-synapses by microglia and astrocytes, thus leading to an increased density of excitatory synapses in the spinal cord after the injury, potentially contributing to the development of chronic pain. While their findings demonstrate a novel synaptic "don't-eat-me" signal that may be potentially targeted to address chronic pain development, the results presented in the manuscript do not support that "selective reorganization" of synapses is observed in the presence of GT1b or that the lack of GT1b "suppresses SNT-induced excitatory synapse remodelling", which are the major conclusions drawn by the authors in this study. Their data clearly demonstrates that there is an increase in the density of excitatory synapses in the spinal cord after the injury and these superfluous synapses are either maintained (in the presence of GT1b) or eliminated to the baseline density (in the absence of GT1b). Therefore, one could state that the lack of GT1b increases network remodelling rather than suppressing it. Furthermore, it is unclear whether in the absence of GT1b remaining synapses are those that were present before the injury or not. Therefore, it is impossible to claim the role of GT1b in selective synaptic remodelling per se beyond its contribution to limiting glial uptake of synapses.

In addition, the relationship between GT1b and synaptic activity is unequivocal, as GT1b-containing synapses have higher frequency, but lower amplitude - the discrepancy that is not discussed in the manuscript.

Minor comments:

1. The description and depiction of attached vs. phagocytosed synaptic particles is confusing. In the first section it is stated: "those in a phagolysosome emitted only the mCherry signal", but later on "entrapped in a phagolysosome (green signal)". In addition, in the figures attached particles are labelled in green rather than yellow, while engulfed are colored in red. Color consistency would improve the readability.
2. Fig. 2D: as there is an increase in total synapse number after the injury, the addition of C3-/GT1b- synapses would increase the transparency of the data.
3. It might be worth separating the section on GT1b vs. synaptic activity and SYK dephosphorylation into two, as these two aspects of network remodelling are not directly related.
4. The concentrations of GT1b used to inhibit phagocytosis and SYK dephosphorylation appear to be quite high. Are they physiologically relevant?
5. Used software should be identified in the methods section "Quantification of spinal synapses".
6. Viral genome number of AAV injections should be identified in the methods section "Synapse-specific pH-reporter expression".

Overall the manuscript is well written and easy to follow. Relevant literature is referenced. The methods are described in sufficient detail.

Referee #2:

Lee et al. analyzed the reorganization of synapses following peripheral nerve injury in mice using a pH-reporter system. Lee et

al. observed that nerve injury triggered an accumulation of the ganglioside GT1b at afferent terminals, which acted protectively against synapse elimination. The inhibition of GT1b-synthesis led to increased glial phagocytosis of excitatory pre-synapses and a subsequent reduction in excitatory synapses post-injury. Lee et al suggest that GT1b-mediated suppression of glial phagocytosis through SYK dephosphorylation.

Data are novel and highly interesting. Results demonstrate a crucial role of the ganglioside GT1b in preventing elimination of synapses after nerve injury. Data give novel insights in the molecular cues that stabilizes synapses against glial phagocytosis after nerve injury.

While the results are well presented and carefully discussed, I would like to kindly suggest a few points for refinement:

- Data show that the ganglioside GT1b acts protective against injury related synapse elimination. However, the current findings do not provide sufficient evidence to conclude that complement is involved in this synapse elimination. Complement factor 3 is only associated with the injury related synapse elimination. Please avoid any misinterpretation of the data and rephrase the following sentence of the abstract 'GT1b acts as a protective signal against complement-mediated synapse elimination'.
- To enhance the clarity and accuracy of the claim in the abstract related to the 'GT1b's pivotal role in selective synapse elimination after nerve injury', please consider to describe the major finding more directly e.g. 'GT1b's pivotal role in preventing synapse elimination after nerve injury'. The same is true for the title. To make the main outcome more explicit, you might consider a better title such as 'Ganglioside GT1b prevents selective spinal synapse removal following peripheral nerve injury'.
- In Figure 1-L. Why is there no comparison conducted between GT1b (-) and GT1b (+) along with marking any significance, since you claim that the microglia preferentially removed GT1b(-) synapses over GT1b(+) synapses
- Figures 1-4. Please give the number of independent samples analyzed in figures. Is each dot in the graphs an independent sample?
- Figure 3. Please define all abbreviations used in the figure e.g. dKO, ipsi, cont, contra, cre, veh etc.
- Figure 4. Please remove the schematic drawing of your hypothesis in figure 4 O from the results part and describe it together with your Expanded View Figure 9.
- Expanded View Figure 9. Graphical summary. The labellings 'Siglec(?)' and 'SHP activation(?)' might not be the most appropriate for a scientific publication. Please could you consider alternative labels, e.g. 'receptor' and 'inhibitory signalling'?

Referee #3:

The authors expand on their previous work to show the influence of Gtb1 within the spinal cord following nerve injury and its role in neuropathic pain. This is a very comprehensive study with in vivo and in vitro elements and adds to the growing literature on glial engulfment of synapses within the spinal cord following nerve injury and consequent synaptic restructuring within the dorsal horn that contributes to long-term changes in nociceptive circuitry that mediate the behavioural effects of peripheral nerve injury. While the techniques used are impressive and the results convincing, there is some issue with the rationale. A previous study (Yousefpour et al 2023) showed that peripheral nerve injury leads to engulfment of both spinal central synapses and presynaptic primary afferent synapses. These can be delineated through their expression of VGLUT2 and VGLUT1 respectively. This study can not distinguish excitatory presynaptic afferents from excitatory presynaptic dorsal horn neurons. Furthermore, Yousefpour et al and Xu et al 2023 both showed inhibitory central synapse engulfment. I am concerned that the use of GAD67 is not sufficient to detect all inhibitory synapses, given it is restricted to gabergic neurons. A more appropriate marker would have been VGAT, expressed by both gabaergic and glycinergic neurons. Yousefpour et al described a distinct temporal pattern of inhibitory and excitatory synaptic engulfment, and Xu et al also showed an increase in inhibitory engulfment after peripheral injury. Given the shift toward net excitation in the dorsal horn following peripheral nerve injury, a decrease in inhibitory tone through selective inhibitory synapse loss seems a likely scenario and is supported by the two studies noted above. This must be addressed in this manuscript - a lack of evidence for microglial inhibitory synapse engulfment is not evidence of it not being apparent - more careful scrutiny through better markers would resolve this.

Cross-comments from referee 1:

I believe that the major reason, why there are just a few experiments suggested is the notations by both reviewers #1 and #2 that the data is overinterpreted. The paper should be reformatted to be better aligned to the experimental findings: GT1b is necessary for the removal of superfluous synapses after injury (rather than mediates „selective reorganization“), C3 is associated with injury-related synapse elimination (but it is not clear whether GT1b acts as a protective signal counteracting C3 specifically) etc.

I think it would be crucial for the reviewers to re-read the paper once these crucial changes are made.

Regarding the experiment suggested by reviewer #3, I believe it should be possible to obtain the requested data using the

leftover tissue from the presented study. I agree that the use of VGAT would deliver more comprehensive results and strengthen the argument provided by the authors. However, if you decide that additional experiments are not essential or if authors cannot deliver them in a timely manner, the limitations outlined by reviewer #3 should be carefully discussed in the manuscript.

Cross-comments from referee 2:

Reviewer #3 is suggesting that "this study cannot distinguish excitatory presynaptic afferents from excitatory presynaptic dorsal horn neurons".

This highly selective topic on neuron/synapse subpopulations is not in my field of expertise.

I think that this comment of reviewer #3 is a more mechanistic follow-up question that does not affect the main claim of the study, namely that "nerve injury triggers a selective reorganization of excitatory synapses that is influenced by the accumulation of the ganglioside GT1b".

However, authors of this manuscript carefully should check all their claims in respect to the criticism made by reviewer #3.

Dear Dr. Schnapp,

Thank you very much for the opportunity to revise our manuscript, entitled “GT1b-mediated selective spinal synapse reorganization following peripheral nerve injury”. We also greatly appreciate the reviewers’ helpful comments and suggestions. We have performed additional experiments and revised the manuscript accordingly. We hope that the changes to our revised manuscript have addressed all the reviewers’ concerns, and that you will find it suitable for publication in *Embo reports*.

We thank you for your review of this manuscript and hope to hear from you soon regarding your decision for publication.

Sincerely,

Sung Joong Lee, Ph.D.
Department of Neuroscience and Physiology
Seoul National University School of Dentistry
Seoul, Republic of Korea
E-mail: sjlee87@snu.ac.kr
Tel: +82-2-880-2309

Response letter

Referee #1

The manuscript by Lee et al. investigates the role of ganglioside GT1b in glia-dependent synapse elimination after spinal cord injury. The authors demonstrate that GT1b inhibits the uptake of excitatory pre-synapses by microglia and astrocytes, thus leading to an increased density of excitatory synapses in the spinal cord after the injury, potentially contributing to the development of chronic pain. While their findings demonstrate a novel synaptic "don't-eat-me" signal that may be potentially targeted to address chronic pain development, the results presented in the manuscript do not support that "selective reorganization" of synapses is observed in the presence of GT1b or that the lack of GT1b "suppresses SNT-induced excitatory synapse remodelling", which are the major conclusions drawn by the authors in this study. Their data clearly demonstrates that there is an increase in the density of excitatory synapses in the spinal cord after the injury and these superfluous synapses are either maintained (in the presence of GT1b) or eliminated to the baseline density (in the absence of GT1b). Therefore, one could state that the lack of GT1b increases network remodelling rather than suppressing it. Furthermore, it is unclear whether in the absence of GT1b remaining synapses are those that were present before the injury or not. Therefore, it is impossible to claim the role of GT1b in selective synaptic remodelling per se beyond its contribution to limiting glial uptake of synapses.

We greatly appreciate this valuable feedback, which has helped us refine the interpretation of our results. We agree with the reviewer that our data may overinterpret the role of GT1b as it relates to "selective reorganization" of synapses. To address this concern, we have made the following revision to the manuscript. First, we have removed the term "selective" from the text to avoid overstatement. Second, we have replaced "synaptic remodeling" with "synaptic expansion" in the Results section related to Figure 3, to more accurately reflect the phenomenon observed in our study. Specifically, we now describe "the inhibition of GT1b synthesis enhances glial phagocytosis and disrupt excitatory synaptic expansion", which better aligns with our findings. These adjustments ensure our conclusions are consistent with the evidence provided and address the reviewer's concerns comprehensively.

In addition, the relationship between GT1b and synaptic activity is unequivocal, as GT1b-containing synapses have higher frequency, but lower amplitude - the discrepancy that is not discussed in the manuscript.

We appreciate this insightful comment. In a previous study by Jafari et al., it was suggested that

activated microglia or invading macrophages mediate the phagocytosis of specific synapses exhibiting higher calcium accumulations[1]. Building on this, our study shows that the pre-synapses with higher calcium amplitude are GT1b-negative, which we interpret as making them more vulnerable to glial phagocytosis.

Although the precise mechanism linking calcium activity to GT1b accumulation remains unclear, we propose that the lack of GT1b facilitates synaptic removal, thereby enhancing network remodeling. To address this point, we have revised the manuscript to include these interpretations and provide a more comprehensive discussion of the observed discrepancies in synaptic frequency and amplitude in the Discussion section (page 14, line 10).

Minor comments:

1. The description and depiction of attached vs. phagocytosed synaptic particles is confusing. In the first section it is stated: "those in a phagolysosome emitted only the mCherry signal", but later on "entrapped in a phagolysosome (green signal)". In addition, in the figures attached particles are labelled in green rather than yellow, while engulfed are colored in red. Color consistency would improve the readability.

We appreciate the reviewer's valuable feedback regarding the description and depiction of attached versus phagocytosed synaptic particles. To improve clarity and consistency, we have revised the manuscript as follow:

"Immunohistochemistry with microglia (Iba-1) and astrocyte (GFAP) markers, followed by 3D reconstruction using IMARIS, categorized the synaptic compartments as either attached/engulfed (yellow signal, representing green puncta in the 3D reconstructed image; herein referred to as attached) or entrapped in a phagolysosome (Red signal, representing red puncta in the 3D reconstructed image; herein referred to as phagocytosed or eliminated) by microglia or astrocytes (Video 1, Fig 1C, and Fig EV1B-E)."

We have also ensured that color labeling is consistent throughout the figures and legends.

2. Fig. 2D: as there is an increase in total synapse number after the injury, the addition of C3-/GT1b-synapses would increase the transparency of the data.

We appreciate the reviewer's valuable suggestion. In response, we have revised the data in Fig. 2D to

include both C3- and GT1b-negative synapses. This addition provides a more transparent and comprehensive representation of the synapse populations after the injury.

3. It might be worth separating the section on GT1b vs. synaptic activity and SYK dephosphorylation into two, as these two aspects of network remodelling are not directly related.

We appreciate the reviewer's suggestion. We have separated the section on GT1b versus synaptic activity and SYK dephosphorylation into 2 distinct sections, as these aspects of network remodeling are not directly related. Correspondingly, we have divided the original figure into Figure 4 and Figure 5 to enhance clarity and focus, as suggested.

4. The concentrations of GT1b used to inhibit phagocytosis and SYK dephosphorylation appear to be quite high. Are they physiologically relevant?

We appreciate the reviewer's concern regarding the concentrations of GT1b used in our experiments. To address the question of physiological relevance, we based our estimation of the physiological concentration of GT1b on available data regarding total ganglioside content in the brain. The brain contains approximately **1–2 $\mu\text{mol/g wet weight}$** of total gangliosides, of which GT1b is reported to comprise around **10–20%** (Schwarz and Futerman, 1996; Yu et al., 2011). Given the molecular weight of GT1b ($\sim 2124 \text{ g/mol}$), this translates to a physiological concentration in the range of **1–2 $\mu\text{g/mL}$** in the tissue environment.

In our experiments, we used **10 $\mu\text{g/mL}$** , a concentration higher than the estimated physiological level, to account for the inherent differences between in vivo and in vitro conditions. In vitro systems lack the complex microenvironment and trafficking dynamics present in vivo, such as lipid bilayer interactions and extracellular matrix components, which can limit the bioavailability and interaction of gangliosides with cells. The higher concentration compensates for these limitations, ensuring sufficient interaction with the cell membrane and allowing for measurable effects.

We also tested a concentration of **1 $\mu\text{g/mL}$** , which falls within the estimated physiological range in the tissue environment. The effects observed at this concentration were consistent with those observed in the **10 $\mu\text{g/mL}$** treated group, though their magnitude was reduced (Fig. 5 C, D). This aligns with the expected dose-dependent relationship.

This justification highlights the balance between experimental feasibility and physiological relevance, and we hope it clarifies the rationale for the concentrations used in this study.

5. Used software should be identified in the methods section "Quantification of spinal synapses".

We appreciate the reviewer's comments. We have included the details of the software used for quantification in the Methods section under "Quantification of spinal synapses". Specifically, we used Image J software with the Synbot and Puncta Analyzer plugins for analysis[2]. This information has been added to ensure transparency and reproducibility (page 22, line 15).

6. Viral genome number of AAV injections should be identified in the methods section "Synapse-specific pH-reporter expression".

We have included the viral genome details in the methods section under "Synapse-specific pH-reporter expression." Specifically, we used AAV2-retro for Ex-Pre and AAV9 for other types of pH-reporters. This information has been added to provide clarity and ensure reproducibility (page 24, line 2-4).

Overall the manuscript is well written and easy to follow. Relevant literature is referenced. The methods are described in sufficient detail.

Referee #2:

Lee et al. analyzed the reorganization of synapses following peripheral nerve injury in mice using a pH-reporter system. Lee et al. observed that nerve injury triggered an accumulation of the ganglioside GT1b at afferent terminals, which acted protectively against synapse elimination. The inhibition of GT1b-synthesis led to increased glial phagocytosis of excitatory pre-synapses and a subsequent reduction in excitatory synapses post-injury. Lee et al suggest that GT1b-mediated suppression of glial phagocytosis through SYK dephosphorylation.

Data are novel and highly interesting. Results demonstrate a crucial role of the ganglioside GT1b in preventing elimination of synapses after nerve injury. Data give novel insights in the molecular cues that stabilizes synapses against glial phagocytosis after nerve injury.

While the results are well presented and carefully discussed, I would like to kindly suggest a few points

for refinement:

- Data show that the ganglioside GT1b acts protective against injury related synapse elimination. However, the current findings do not provide sufficient evidence to conclude that complement is involved in this synapse elimination. Complement factor 3 is only associated with the injury related synapse elimination. Please avoid any misinterpretation of the data and rephrase the following sentence of the abstract 'GT1b acts as a protective signal against complement-mediated synapse elimination'.

We appreciate the reviewer's valuable feedback regarding the interpretation of our data. To avoid any potential misinterpretation, we have revised the sentence in the abstract as follows: "GT1b acts as a protective signal against nerve injury-induced spinal synapse elimination." This change ensures that the abstract accurately reflects the findings without overstating the role of complement in synapse elimination, aligning with the presented data.

- To enhance the clarity and accuracy of the claim in the abstract related to the 'GT1b's pivotal role in selective synapse elimination after nerve injury', please consider to describe the major finding more directly e.g. 'GT1b's pivotal role in preventing synapse elimination after nerve injury'. The same is true for the title. To make the main outcome more explicit, you might consider a better title such as 'Ganglioside GT1b prevents selective spinal synapse removal following peripheral nerve injury'.

We appreciate the reviewer's valuable suggestion to enhance the clarity and accuracy of the abstract and title. We have revised both as recommended. The abstract now directly describes the major finding, and the title has been updated to: "Ganglioside GT1b prevents selective spinal synapse removal following peripheral nerve injury." These changes make the main outcome more explicit and align with the core findings of our study.

- In Figure 1-L. Why is there no comparison conducted between GT1b (-) and GT1b (+) along with marking any significance, since you claim that the microglia preferentially removed GT1b(-) synapses over GT1b(+) synapses

We appreciate the reviewer's comment. We have revised the Results section to clarify our findings. Specifically, we updated the statement from "we found that the microglia preferentially removed GT1b negative synapses over GT1b positive synapses" to "we found that the microglial phagocytosis of GT1b-negative synapses increased significantly after injury." (page 7, line 24). This revision addresses the reviewer's concern and ensures the data presentation aligns with our claims.

- Figures 1-4. Please give the number of independent samples analyzed in figures. Is each dot in the graphs an independent sample?

We appreciate the reviewer's valuable comment. Each dot in the graphs represents an independent sample, and this information has been included in the data source files. Additionally, some data points represent the average value derived from multiple slides obtained from a single mouse.

- Figure 3. Please define all abbreviations used in the figure e.g. dKO, ipsi, cont, contra, cre, veh etc.

We have added definitions for all abbreviations used in Figure 3 to the figure legend for clarity. The updated legend now includes the following:

Abbreviations: Veh, vehicle control for PDMP; Ipsi, ipsilateral side; Contra, contralateral side; SNT, L5 spinal nerve transection model for neuropathic pain; Sham, sham control for SNT; dKO, double KO of St3gal2/3; Cre, Scn10a-Cre.

- Figure 4. Please remove the schematic drawing of your hypothesis in figure 4 O from the results part and describe it together with your Expanded View Figure 9.

- Expanded View Figure 9. Graphical summary. The labellings 'Siglec(?)' and 'SHP activation(?)' might not be the most appropriate for a scientific publication. Please could you consider alternative labels, e.g. 'receptor' and 'inhibitory signalling'?

We have made the suggested changes. The schematic drawing of our hypothesis, previously included in Figure 4O, has been removed from the Results section and is now described together with the Graphical Summary (Expanded view Figure 9). Additionally, the labels in the Graphical Summary have been revised for clarity and appropriateness, replacing “Siglec9(?)” and “SHP activation(?)” with “Receptor” and “Inhibitory Signaling,” respectively, to ensure suitability for a scientific publication.

Referee #3:

The authors expand on their previous work to show the influence of Gtb1 within the spinal cord

following nerve injury and its role in neuropathic pain. This is a very comprehensive study with in vivo and in vitro elements and adds to the growing literature on glial engulfment of synapses within the spinal cord following nerve injury and consequent synaptic restructuring within the dorsal horn that contributes to long-term changes in nociceptive circuitry that mediate the behavioural effects of peripheral nerve injury. While the techniques used are impressive and the results convincing, there is some issue with the rationale. A previous study (Yousefpour et al 2023) showed that peripheral nerve injury leads to engulfment of both spinal central synapses and presynaptic primary afferent synapses. These can be delineated through their expression of VGLUT2 and VGLUT1 respectively. This study can not distinguish excitatory presynaptic afferents from excitatory presynaptic dorsal horn neurons. Furthermore, Yousefpour et al and Xu et al 2023 both showed inhibitory central synapse engulfment. I am concerned that the use of GAD67 is not sufficient to detect all inhibitory synapses, given it is restricted to gabaergic neurons. A more appropriate marker would have been VGAT, expressed by both gabaergic and glycinergic neurons. Yousefpour et al described a distinct temporal pattern of inhibitory and excitatory synaptic engulfment, and Xu et al also showed an increase in inhibitory engulfment after peripheral injury. Given the shift toward net excitation in the dorsal horn following peripheral nerve injury, a decrease in inhibitory tone through selective inhibitory synapse loss seems a likely scenario and is supported by the two studies noted above. This must be addressed in this manuscript - a lack of evidence for microglial inhibitory synapse engulfment is not evidence of it not being apparent - more careful scrutiny through better markers would resolve this.

We thank reviewer for the valuable comments and insightful suggestions, which have greatly improved the scope and direction of our manuscript.

We acknowledge the concern regarding the inability to distinguish between excitatory presynaptic afferents and excitatory presynaptic dorsal horn neurons. This limitation is recognized in our study. Our primary focus was on primary afferent terminals, as our previous work demonstrated that peripheral nerve injury induces GT1b production in DRG neurons,

which is subsequently transported and accumulates in presynaptic afferent terminals, as validated through rhizotomy experiments[3]. However, we agree with the reviewer that VGLUT2 and excitatory presynaptic pH reporter reflect not only afferent terminals but also excitatory pre-synapses of spinal interneurons. To address this point, we will revise the manuscript to clarify this distinction, modifying references to “afferent terminals” to “pre-synapses in the superficial layer of spinal dorsal horn.”

We used GAD as a promoter for the inhibitory pre-synaptic pH reporter based on the observation that the most integrated presynaptic terminals surrounding pain projection neurons are GABAergic, whereas glycinergic neuronal terminals are predominantly located in deeper layers (below L III) and are less associated with pain projection neurons[4]. Therefore, GAD was selected as a promoter to optimally visualize inhibitory presynaptic terminals integrated with pain projection neurons.

We appreciate the reviewer’s concern that GAD67 alone may not be sufficient to detect all inhibitory synapses, particularly those originating from glycinergic neurons. To address this, we conducted additional analysis using VGAT and CD68 markers alongside Iba-1 and GFAP to assess microglial and astrocytic phagocytosis of inhibitory pre-synapses, thereby encompassing both GABAergic and glycinergic synapses. Our data indicate that neither microglia nor astrocytes phagocytosed inhibitory pre-synapses labeled by VGAT. We have included this additional data as Expanded View Figure 3 to support our findings.

We appreciate the reviewer’s reference to the findings of Xu et al[5]. and Yousefpour et al[6]., which provide valuable contextual insights. Xu et al. utilized a hind-paw incision model and observed mild microglial activation, accompanied by significant increases in the phagocytosis of both VGLUT2+ excitatory and VGAT+ inhibitory pre-synapses, with a preference for inhibitory pre-synapses. Yousefpour et al. employed a cuffing mouse model (PNI), reporting

microglial activation primarily in the deeper layer of the spinal dorsal horn (below L III) and mechanical allodynia 10 days post-injury. IHC samples collected 20 days post-PNI revealed significant increase in both VGLUT2+ excitatory pre-synapses and VGAT+ inhibitory pre-synapses at comparable levels, albeit with a distinct temporal pattern.

In contrast, our study utilized the L5 spinal nerve transection (L5 SNT) model, which is characterized by immediate microglial activation in the superficial layer, an immediate pain response (allodynia and hyperalgesia), and IHC samples collected at 3- and 7-days post-injury. Focusing on the outer region of laminae I-II, where pain projection neurons are located, we observed significant phagocytosis of excitatory presynaptic pH reporters expressed under the CaMKIIa promoter following L5 SNT. These findings suggest that differences in the injury model, the onset of pain behavior, and the spinal cord regions examined may contribute to variations in the structural reorganization pattern.

We acknowledge that our study may not have fully captured inhibitory synapse elimination, particularly during the later post-injury phase or in the deeper layers of the spinal dorsal horn. We incorporated these insights into our manuscript to provide more comprehensive perspective on the synaptic alterations associated with neuropathic pain (page 13, line 14). We sincerely thank the reviewer for their valuable and constructive feedback.

Cross-comments from referee 1:

I believe that the major reason, why there are just a few experiments suggested is the notations by both reviewers #1 and #2 that the data is overinterpreted. The paper should be reformatted to be better aligned to the experimental findings: GT1b is necessary for the removal of superfluous synapses after injury

(rather than mediates „selective reorganization"), C3 is associated with injury-related synapse elimination (but it is not clear whether GT1b acts as a protective signal counteracting C3 specifically) etc.

I think it would be crucial for the reviewers to re-read the paper once these crucial changes are made.

Regarding the experiment suggested by reviewer #3, I believe it should be possible to obtain the requested data using the leftover tissue from the presented study. I agree that the use of VGAT would deliver more comprehensive results and strengthen the argument provided by the authors. However, if you decide that additional experiments are not essential or if authors cannot deliver them in a timely manner, the limitations outlined by reviewer #3 should be carefully discussed in the manuscript.

We appreciate the valuable comments. we performed an experiment referee asked using VGAT which is marker for both glycinergic and GABAergic pre-synapses, and the results showed that there was no differences in phagocytosis of inhibitory pre-synapses (Fig EV4).

Cross-comments from referee 2:

Reviewer #3 is suggesting that "this study cannot distinguish excitatory presynaptic afferents from excitatory presynaptic dorsal horn neurons".

This highly selective topic on neuron/synapse subpopulations is not in my field of expertise.

I think that this comment of reviewer #3 is a more mechanistic follow-up question that does not affect the main claim of the study, namely that "nerve injury triggers a selective reorganization of excitatory synapses that is influenced by the accumulation of the ganglioside GT1b".

However, authors of this manuscript carefully should check all their claims in respect to the criticism made by reviewer #3.

We appreciate the valuable comments.

1. Jafari M, Schumacher AM, Snaidero N, Ullrich Gavilanes EM, Neziraj T, Kocsis-Jutka V, et al. Phagocyte-mediated synapse removal in cortical neuroinflammation is promoted by local calcium

accumulation. *Nat Neurosci.* 2021;24(3):355-67. Epub 20210125. doi: 10.1038/s41593-020-00780-7. PubMed PMID: 33495636.

2. Savage JT, Ramirez JJ, Risher WC, Wang Y, Irala D, Eroglu C. SynBot is an open-source image analysis software for automated quantification of synapses. *Cell Reports Methods.* 2024;4(9):100861. doi: <https://doi.org/10.1016/j.crmeth.2024.100861>.

3. Lim H, Lee J, You B, Oh JH, Mok HJ, Kim YS, et al. GT1b functions as a novel endogenous agonist of toll-like receptor 2 inducing neuropathic pain. *Embo j.* 2020;39(6):e102214. Epub 20200206. doi: 10.15252/emboj.2019102214. PubMed PMID: 32030804; PubMed Central PMCID: PMC7073460.

4. Shimizu-Okabe C, Kobayashi S, Kim J, Kosaka Y, Sunagawa M, Okabe A, et al. Developmental Formation of the GABAergic and Glycinergic Networks in the Mouse Spinal Cord. *International Journal of Molecular Sciences.* 2022;23(2):834. PubMed PMID: doi:10.3390/ijms23020834.

5. Xu Y, Moulding D, Jin W, Beggs S. Microglial phagocytosis mediates long-term restructuring of spinal GABAergic circuits following early life injury. *Brain, Behavior, and Immunity.* 2023;111:127-37. doi: <https://doi.org/10.1016/j.bbi.2023.04.001>.

6. Yousefpour N, Locke S, Deamond H, Wang C, Marques L, St-Louis M, et al. Time-dependent and selective microglia-mediated removal of spinal synapses in neuropathic pain. *Cell Rep.* 2023;42(1):112010. Epub 20230118. doi: 10.1016/j.celrep.2023.112010. PubMed PMID: 36656715.

Dear Prof. Lee,

Thank you for the submission of your revised manuscript. We have now received the enclosed report from referee 2 who was asked to assess it. Referee 1 was unfortunately not available to re-review your study and I apologize for our delayed response, due to our repeated trials to reach out to referee 1.

I am happy to say that referee 2 supports the publication of your revised ms and that only a few more minor changes are required before we can proceed with the official acceptance of your work. Please address all comments by referee 2.

In addition:

- Your ms has 5 main figures and will thus be published as a scientific report with combined results and discussion sections. Please label this section as such.
- The Data Availability Section (DAS) has incorrect content that needs to be removed (Lead contact, Materials availability, Data and code availability). The DAS should only list data that are deposited in public databases. If you have not deposited any data, please mention this fact in the DAS. Also, the DAS needs to be placed before the Acknowledgments.
- There are some author name discrepancies - Subeen Lee in the ms vs. Subin Lee in our online submission system; Sungho Chang in the ms vs. Sunghoe Chang online. Please correct.
- The author credits need to be removed from the ms file, all credits need to be entered during online ms submission.
- The REFERENCE format needs to be corrected: it needs to be alphabetical, not numerical; et al needs to be used after 10 author names; DOIs should only be used for preprints and datasets that have not been published yet. Please use the EMBO reports reference style.
- The correct nomenclature for the movie files is Movie EV1 and Movie EV2; the legends need to be removed from the ms and each should be provided in a readme.txt file and then zipped up with its movie file and uploaded as folder per movie; the source file names, titles, legends and zip folders need to be updated to Movie EV1 and Movie EV2.
- The Reagent & Tools table file is only needed as a separate file. Please remove it from the ms.
- The nomenclature for EV figure legends in the ms is not correct: it should be Figure EV1, instead of Expanded View Figure 1.
- Materials and methods should be just Methods

During our standard image analysis, our Data Integrity analyst detected potential aberrations in the figure set, and we would like to clarify these issues before we can proceed with your paper here. We kindly invite you to check the composition of: Figure 3D. Contains cell images with distinct repeating patterns.

Please explain these anomalies in the cell images. If you make changes to the figure set, please include a point-by-point describing what you have changed and why. Please see the attached image for clarification.

Figure Legends - Comments

- Please note that the exact p values are not provided in the legends of figures 1D, F, I, L, N; 2B, D; 3C, F, I, L, O; 4D, 5D, F, H, J; EV1 C, D, E; EV3 B, D; EV5 B, EV6 A, B
- Please note that in figures 4D, EV1 C, D, E; EV3 B, D; EV5 B, EV6 A, B there is a mismatch between the annotated p values in the figure legend and the annotated p values in the figure file that should be corrected.
- Please indicate what */ **/ ***/ **** represents; if this represents p value(s), please indicate the statistical test used and where appropriate, specify the exact p value in the legend(s) of figure(s) EV2 B, C
- Please note that information related to n is missing in the legends of figures 1D, E, F, G, I, L, N; 2B, D; 3C, F, H, I, L, O; 4D, 5D, F, H, J; EV1 B, C, D, E; EV2 B, C; EV3 B, D, F; EV4 B, D; EV5 B, EV6 A, B.
- Please note that the error bars are not defined in the legends of figures EV2 B, C.
- Please note that the scale bar needs to be defined for figures EV4 C
- Please note that scale bar and its definition are missing for figures 2E, 3K, N; EV4 A, EV5 A
- Please note that the dotted border is not defined in the legends of figures 2C, 4C. This needs to be rectified.
- Please note that the yellow arrow heads are not defined in the legend of figure 3C. This needs to be rectified.

The exact p-values should always be provided, as reasonable.

I would like to suggest a few minor changes to the abstract that needs to be written in present tense. Please let me know whether you agree with the following:

After peripheral nerve injury, the structure of the spinal cord is actively regulated by glial cells, contributing to the chronicity of neuropathic pain. However, the mechanism by which peripheral nerve injury leads to synaptic imbalance remains elusive. Here, we use a pH-reporter system and find that nerve injury triggers a reorganization of excitatory synapses that is influenced by the accumulation of the ganglioside GT1b at afferent terminals. GT1b acts as a protective signal against nerve injury-induced spinal synapse elimination. Inhibition of GT1b-synthesis increases glial phagocytosis of excitatory pre-synapses and reduces excitatory synapses post-injury. In vitro analyses reveal a positive correlation between GT1b accumulation and the frequency of pre-synaptic calcium activity, with GT1b-mediated suppression of glial phagocytosis occurring through SYK dephosphorylation. Our study highlights GT1b's pivotal role in preventing synapse elimination after nerve injury and offers new insight into the molecular underpinning of activity-dependent synaptic stability and glial phagocytosis.

EMBO press papers are accompanied online by A) a short (1-2 sentences) summary of the findings and their significance, B) 2-3 bullet points highlighting key results and C) a synopsis image that is exactly 550 pixels wide and 200-600 pixels high (the height is variable). The synopsis image should provide a sketch of the major findings, like a graphical abstract. Please note that text needs to be readable at the final size. Please send us this information along with the final manuscript.

Referee #2:

Lee et al carefully revised the manuscript. The overinterpretations of the previous manuscript version have been removed and adjusted. In addition, Lee et al performed additional experiments that were requested by reviewer #3. The now show by using VGAT as a marker for both glycinergic and GABAergic pre-synapses that there was no difference in phagocytosis of inhibitory pre-synapses after peripheral nerve injury (see Fig EV4). This additional novel finding clearly strengthens the claims that „GT1b acts as a protective signal against nerve injury-induced spinal synapse elimination" and that „inhibition of GT1b-synthesis led to increased glial phagocytosis of excitatory pre-synapses".

The data are novel and interesting. They demonstrate a crucial role of the ganglioside GT1b in protection against glial phagocytosis after peripheral nerve injury.

There are still some minor points in the figures that should be corrected or improved:

- Figure 1 C: Labelling 'Microglia' and 'Astrocyte' are only partially visible
- Figure 4 I: Please also add a labelling to the y-axis
- Figure 4 B and Figure 5 B: Character size of the text in the schematic drawings is very small
- Expanded View Figure 4 B: Labelling of scale bar is missing
- Expanded View Figure 5 A: Please add a scale bar

Dear Dr. Schnapp,

Thank you for the positive evaluation of our revised manuscript. We have made the necessary revisions based on your comments as well as those from the reviewer #2. We believe that the changes we've made address all the feedback, and hope that you will find it suitable for publication in *Embo reports*.

We appreciate your time and effort in reviewing this manuscript, and look forward to hearing your decision regarding this publication.

Sincerely,

Sung Joong Lee, Ph.D.
Department of Neuroscience and Physiology
Seoul National University School of Dentistry
Seoul, Republic of Korea
E-mail: sjlee87@snu.ac.kr
Tel: +82-2-880-2309

Editor's comments:

- Your ms has 5 main figures and will thus be published as a scientific report with combined results and discussion sections. Please label this section as such.

we appreciate the editor's comment. The section has been labeled as requested.

- The Data Availability Section (DAS) has incorrect content that needs to be removed (Lead contact, Materials availability, Data and code availability). The DAS should only list data that are deposited in public databases. If you have not deposited any data, please mention this fact in the DAS. Also, the DAS needs to be placed before the Acknowledgments.

We appreciate the editor's comment. We have removed the unnecessary information from the Data Availability Section (DAS) and clearly stated that no data have been deposited in public databases. Additionally, the DAS has been moved to appear before the acknowledgements section as requested.

- There are some author name discrepancies - Subeen Lee in the ms vs. Subin Lee in our online submission system; Sungho Chang in the ms vs. Sunghoe Chang online. Please correct.

We have corrected the author names to ensure consistency between the manuscript and the online submission system.

- The author credits need to be removed from the ms file, all credits need to be entered during online ms submission.

Author credits have been removed from the manuscript file and properly entered during online

manuscript submission as requested.

- The REFERENCE format needs to be corrected: it needs to be alphabetical, not numerical; et al needs to be used after 10 author names; DOIs should only be used for preprints and datasets that have not been published yet. Please use the EMBO reports reference style.

The reference list has been reformatted according to the EMBO reports style.

- The correct nomenclature for the movie files is Movie EV1 and Movie EV2; the legends need to be removed from the ms and each should be provided in a readme.txt file and then zipped up with its movie file and uploaded as folder per movie; the source file names, titles, legends and zip folders need to be updated to Movie EV1 and Movie EV2.

We have corrected the nomenclature to Movie EV1 and Movie EV2, removed the movie legends from the manuscript, and provided each legend separately in a txt file. Each movie file, along with its txt file, has been zipped and uploaded as a separate folder, as instructed.

- The Reagent & Tools table file is only needed as a separate file. Please remove it from the ms.

The reagents & tools table has now been removed from the manuscript and provided as a separate file, as instructed.

- The nomenclature for EV figure legends in the ms is not correct: it should be Figure EV1, instead of Expanded View Figure 1.

The figure legends have been corrected to the appropriate nomenclature as instructed.

- Materials and methods should be just Methods

The section title has been updated to “methods” as instructed.

During our standard image analysis, our Data Integrity analyst detected potential aberrations in the figure set, and we would like to clarify these issues before we can proceed with your paper here. We kindly invite you to check the composition of:

Figure 3D. Contains cell images with distinct repeating patterns.

Please explain these anomalies in the cell images. If you make changes to the figure set, please include a point-by-point describing what you have changed and why. Please see the attached image for clarification.

We Sincerely apologize for submitting the incorrect figure. In the original manuscript, Figure 3D initially showed representative images of the ipsilateral spinal cord only. However, during our internal revision process, we concluded that including representative images of the contralateral spinal cord would enhance clarity and understanding. Thus, we prepared example images for internal comparison purposes. Unfortunately, during figure rearrangement and manuscript editing, these temporary example images were mistakenly incorporated into the final submission.

This oversight was entirely unintentional and reflects an error in our internal editing procedure rather than any attempt to provide misleading data. We deeply regret this mistake and have now replaced Figure 3D with the correct and verified representative images. Additionally, we have carefully verified all other figures in the manuscript to ensure accuracy.

Figure Legends - Comments

- Please note that the exact p values are not provided in the legends of figures 1D, F, I, L, N; 2B, D;

3C, F, I, L, O; 4D, 5D, F, H, J; EV1 C, D, E; EV3 B, D; EV5 B, EV6 A, B

We have provided exact p values in the legends of each figure.

- Please note that in figures 4D, EV1 C, D, E; EV3 B, D; EV5 B, EV6 A, B there is a mismatch between the annotated p values in the figure legend and the annotated p values in the figure file that should be corrected.

Mismatched p values annotated within each figure have been corrected to ensure consistency.

- Please indicate what */ **/ ***/ **** represents; if this represents p value(s), please indicate the statistical test used and where appropriate, specify the exact p value in the legend(s) of figure(s) EV2 B, C

We have clearly defined these symbols and indicated the corresponding statistical tests used in the legends of each figure.

- Please note that information related to n is missing in the legends of figures 1D, E, F, G, I, L, N; 2B, D; 3C, F, H, I, L, O; 4D, 5D, F, H, J; EV1 B, C, D, E; EV2 B, C; EV3 B, D, F; EV4 B, D; EV5 B, EV6 A, B.

We have provided detailed information regarding n, sample size in all relevant figure legends.

- Please note that the error bars are not defined in the legends of figures EV2 B, C.

The error bars presented in Figure EV2 have now been clearly defined within their legends.

- Please note that the scale bar needs to be defined for figures EV4 C

- Please note that scale bar and its definition are missing for figures 2E, 3K, N; EV4 A, EV5 A

We have defined the scale bars explicitly in each figure. For Figure 3K and 3N, as they represent colocalized images derived from Figure 3J and 3L, respectively, the scale bars are identical and now explicitly stated in the legend.

- Please note that the dotted border is not defined in the legends of figures 2C, 4C. This needs to be rectified.

Definitions for dotted borders have been clearly provided in the legends of each figure.

- Please note that the yellow arrow heads are not defined in the legend of figure 3C. This needs to be rectified.

The meaning of yellow arrowheads has been explicitly stated in the legend.

The exact p-values should always be provided, as reasonable.

All statistical analyses and raw data details are provided clearly in the submitted source data file, as instructed.

I would like to suggest a few minor changes to the abstract that needs to be written in present tense. Please let me know whether you agree with the following:

After peripheral nerve injury, the structure of the spinal cord is actively regulated by glial cells, contributing to the chronicity of neuropathic pain. However, the mechanism by which peripheral nerve injury leads to synaptic imbalance remains elusive. Here, we use a pH-reporter system and find that nerve injury triggers a reorganization of excitatory synapses that is influenced by the accumulation of the ganglioside GT1b at afferent terminals. GT1b acts as a protective signal against nerve injury-induced spinal synapse elimination. Inhibition of GT1b-synthesis increases glial phagocytosis of excitatory pre-synapses and reduces excitatory synapses post-injury. In vitro analyses reveal a positive correlation between GT1b accumulation and the frequency of pre-synaptic calcium activity, with GT1b-mediated suppression of glial phagocytosis occurring through SYK dephosphorylation. Our study highlights GT1b's pivotal

role in preventing synapse elimination after nerve injury and offers new insight into the molecular underpinning of activity-dependent synaptic stability and glial phagocytosis.

We appreciate editor's suggested improvements to our abstract. We fully agree with the proposed changes, and the abstract has been revised accordingly, ensuring consistent use of the present tense as recommended.

EMBO press papers are accompanied online by A) a short (1-2 sentences) summary of the findings and their significance, B) 2-3 bullet points highlighting key results and C) a synopsis image that is exactly 550 pixels wide and 200-600 pixels high (the height is variable). The synopsis image should provide a sketch of the major findings, like a graphical abstract. Please note that text needs to be readable at the final size. Please send us this information along with the final manuscript.

A) Short summary

Peripheral nerve injury triggers synaptic remodeling regulated by GT1b, which stabilizes spinal excitatory synapses by suppressing glial phagocytosis. This study uncovers GT1b as a critical protective factor maintaining synaptic integrity following nerve injury.

B) Bullet points

- Peripheral nerve injury selectively induces glial phagocytosis of spinal excitatory pre-synapses.
- Ganglioside GT1b accumulates at synaptic terminal preventing their removal by glial cells following nerve injury.
- GT1b functions as a novel synaptic "don't eat me" signal, suppressing glial phagocytosis through SYK dephosphorylation.

I look forward to seeing a final version of your manuscript as soon as possible. Please use this link to submit your revision: <https://embor.msubmit.net/cgi-bin/main.plex>

Best regards,

Esther

Esther Schnapp, PhD

Senior Editor

EMBO reports

Referee #2:

Lee et al carefully revised the manuscript. The overinterpretations of the previous manuscript version have been removed and adjusted. In addition, Lee et al performed additional experiments that were requested by reviewer #3. The now show by using VGAT as a marker for both glycinergic and GABAergic pre-synapses that there was no difference in phagocytosis of inhibitory pre-synapses after peripheral nerve injury (see Fig EV4). This additional novel finding clearly strengthens the claims that „GT1b acts as a protective signal against nerve injury-induced spinal synapse elimination" and that „inhibition of GT1b-synthesis led to increased glial phagocytosis of excitatory pre-synapses".

The data are novel and interesting. They demonstrate a crucial role of the ganglioside GT1b in protection against glial phagocytosis after peripheral nerve injury.

There are still some minor points in the figures that should be corrected or improved:

- Figure 1 C: Labelling 'Microglia' and 'Astrocyte' are only partially visible
- Figure 4 I: Please also add a labelling to the y-axis
- Figure 4 B and Figure 5 B: Character size of the text in the schematic drawings is very small
- Expanded View Figure 4 B: Labelling of scale bar is missing
- Expanded View Figure 5 A: Please add a scale bar

We sincerely thank Referee #2 for recognizing our revision and additional experiments. We greatly appreciate the minor points raised and have addressed each one carefully. All figure issues have been corrected as instructed.

Dear Prof. Lee,

Thank you for the submission of your revised ms files. They look good overall, however, the explanation for the image duplications in Fig 3D is not very clear. Could you please explain what happened in a little more detail? It seems that all duplicated images are derived from the same single source image? It would be good to provide a better explanation, also for our transparent peer-review file that is by default linked to your paper upon publication.

Also, the text in the synopsis image is a little too small and difficult to read at the final image size. Could you please send us an improved synopsis image with larger text?

The abstract has not been corrected, but if you agree with the suggested changes, we will replace the abstract for you.

Best regards,

Esther

Esther Schnapp, PhD

Senior Editor

EMBO reports

Dear Dr. Schnapp,

Thank you very much for your careful review and feedback. Allow me to clarify in more detail how the duplicated images appeared in Figure 3D of our manuscript.

Previous (original) figure

Initially, we selected ipsilateral representative images to illustrate the differences between the Sham and SNT groups (Previous figure). However, during internal discussions and revisions, our team considered including contralateral images alongside ipsilateral images to provide balanced comparisons.

In the process of identifying appropriate contralateral representative images, we initially found suitable PDMP_Sham_contra images. However, matching each ipsilateral image to the correct corresponding contralateral image took additional time.

Ipsi_Veh_Sham_cropping

Thus, to expedite internal discussions and evaluate the overall layout and visual balance of the main figure, we temporarily created illustrative example images by cropping and duplicating a single ipsi_Veh_Sham image (highlighted in colored boxes within the figure). These examples were intended strictly for internal visualization purposes only.

Previous figure

VS

Temporarily created example image

Due to the extensive raw imaging data contributed by multiple researchers (J Lee, S Lee, and S Chung), this temporary comparative layout was inadvertently incorporated into the Fig. 3D during manuscript preparation. To clarify explicitly, these duplicated images originated unintentionally from a single source image that was used solely as an internal illustrative example.

Upon recognizing this oversight, we immediately replaced the mistakenly included images with the accurate, original contralateral images that have been fully verified against all raw data. We have thoroughly reviewed all figures to confirm no additional errors occurred and assure complete accuracy in our final submission.

Regarding the synopsis image, as per your suggestion, we have now increased the text size for enhanced readability and attached the updated version.

As for the abstract correction, we agree with the replacement as suggested.

If there are any further adjustments required or if you have additional questions, please let us know. We appreciate your diligence in handling our manuscript.

Sung Joong Lee, Ph.D

Prof. Sung Joong Lee
Seoul National University School of Dentistry
Neuroscience and Oral physiology
gwanak-gu, gwanak-ro 1
Seoul 08826
Korea, Republic of

Dear Prof. Lee,

I am very pleased to accept your manuscript for publication in the next available issue of EMBO reports. Thank you for your contribution to our journal.
